# Acoustic ejection mass spectrometry empowers ultra-fast protein biomarker quantification

Bart Van Puyvelde [1,2,8], Christie L. Hunter[3,8], Maxim Zhgamadze[2], Sudha Savant[4], Y. Oliver Wang [2], Esthelle Hoedt[2], Koen Raedschelders[2], Matt Pope[5], Carissa A. Huynh[6], V. Krishnan Ramanujan[6], Warren Tourtellotte[6], Morteza Razavi[5], N. Leigh Anderson[5], Geert Martens [7], Dieter Deforce [1], Qin Fu[2], Maarten Dhaenens [1,9] ✉ & Jennifer E. Van Eyk [2,9] ✉

The global scientific response to COVID 19 highlighted the urgent need for increased throughput and capacity in bioanalytical laboratories, especially for the precise quantification of proteins that pertain to health and disease. Acoustic ejection mass spectrometry (AEMS) represents a much-needed paradigm shift for ultra-fast biomarker screening. Here, a quantitative AEMS assays is presented, employing peptide immunocapture to enrich (i) 10 acute phase response (APR) protein markers from plasma, and (ii) SARS-CoV-2 NCAP peptides from nasopharyngeal swabs. The APR proteins were quantified in 267 plasma samples, in triplicate in 4.8 h, with %CV from 4.2% to 10.5%. SARS-CoV-2 peptides were quantified in triplicate from 145 viral swabs in 10 min. This assay represents a 15-fold speed improvement over LC-MS, with instrument stability demonstrated across 10,000 peptide measurements. The combination of speed from AEMS and selectivity from peptide immunocapture enables ultra-high throughput, reproducible quantitative biomarker screening in very large cohorts.

Epidemiology and population sciences study the patterns, causes, and effects of health and disease conditions within and across populations, and play a pivotal role in guiding policy decisions and advancing medical research. The success of epidemiological studies, particularly within the context of urgent public health crises like COVID-19, is predicated on the efficient and accurate analysis of very large numbers of samples. The development of improved diagnostic and prognostic proteomic markers similarly requires high-throughput methods to analyze sample cohorts that are sufficiently sized to accommodate biological variation. Equally important is the ability to accurately quantify proteins in body fluids for validation studies of potential circulating biomarkers destined for assessment in clinical trials or eventual clinical utilization[1]. This demand for high sample throughput however can collide with the limitations of traditional analytical techniques. In the case of protein quantification from highly complex biological matrices, untargeted proteomics using liquid chromatography-mass spectrometry (LC-MS) is a highly selective, accurate, and routinely utilized platform[2]. Spurred by the demands of the SARS-CoV-2 pandemic, LC-MS discovery proteomics workflows reached unprecedented throughputs[3–5], attaining 1-min-per-sample

[1]ProGenTomics, Laboratory of Pharmaceutical Biotechnology, Ghent University, 9000 Ghent, Belgium. [2]Advanced Clinical Biosystems Research Institute, Smidt Heart Institute, Cedars-Sinai Medical Center, Los Angeles, CA 90048, USA. [3]SCIEX, Redwood City, CA 94065, USA. [4]Beckman Coulter Life Sciences, Brea, CA, USA. [5]SISCAPA Assay Technologies Inc., Box 53309 Washington, DC 20009, USA. [6]Cedars Sinai Biobank & Research Pathology Resource, Cedars-Sinai Medical Center, Los Angeles, CA 90048, USA. [7]AZ Delta Medical Laboratories, AZ Delta General Hospital, 8800 Roeselare, Belgium. [8]These authors contributed equally: Bart Van Puyvelde, Christie L. Hunter. [9]These authors jointly supervised this work: Maarten Dhaenens, Jennifer E. Van Eyk. ✉e-mail: Maarten.Dhaenens@ugent.be; Jennifer.VanEyk@cshs.org

with standardized automated sample preparation[6]. Despite the promise these studies harbor for using discovery proteomics in clinical settings, complex samples still burden the system considerably, compromising robustness on large cohorts, and chromatographic separation remains paramount to the robust identification and quantification of proteins.

COVIDPro recently bundled plasma proteome biomarker discovery efforts in an easily interrogatable database[7]. Among these non-targeted LC-MS studies, a subset of quantified acute phase response (APR) proteins were found to be changing during viral infection[3–5]. While several protein markers of SARS-CoV-2 disease severity show consistent trends, discrepancies underscore a need for targeted screening approaches focused solely on specific biomarkers of interest so that scaled cohorts encompassing tens—to hundreds—of thousands of samples might be within reach[7].

Alongside the numerous protein biomarker studies in plasma, the SARS-CoV-2 pandemic also spurred a concerted drive to develop a protein-based SARS-CoV-2 screening method in nasopharyngeal swabs[8–12]. These efforts were at least partially driven by the challenges of cross contamination, standardization and quantification of RT-qPCR assays across laboratories[13]. Like human plasma, nasopharyngeal swab storage media is a complex matrix that is challenging for the direct quantification of viral peptides[12]. Therefore, an immunoaffinity step, applied during sample preparation, purifies the target peptides to improve the sensitivity of this targeted assay and substantially shorten the LC-MS run to 1–2 min[8,9]. Most recently, a MALDI-TOF approach describes considerably higher throughput, albeit with a compromise in sensitivity[14].

These efforts represent enormous progress, yet the peptide chromatographic separation step in standard LC-MS-based measurements remains a significant constraint to throughput. In this study, automated immunocapture for peptide enrichment has been combined with Acoustic Ejection Mass Spectrometry (AEMS) to yield an ultra-high throughput, LC-free workflow in which a peptide/protein is quantified every 1.5 second (Fig. 1A). AEMS is a cutting-edge technique that achieves very high sampling rates by harnessing acoustic waves to propel 2.5 nL sample droplets into a flow stream for mass spectrometric analysis. The Echo MS system (Fig. 1B), incorporating Acoustic Droplet Ejection (ADE), Open Port Interface (OPI), and a mass spectrometer, exemplifies AEMS technology[15–17]. The technology has primarily been utilized for quantifying small molecules in high-throughput screening applications[18–20]. While AEMS has been explored for intact protein analysis using a prototype configuration with a high-resolution quadrupole time-of-flight (QTOF) device, its potential utility in protein biomarker quantification remained unexplored[21].

The AEMS workflow has no chromatographic separation and lacks desalting, so matrix complexity and non-specific contaminants can compromise its effectiveness. The analysis of digested proteome samples derived from highly complex biofluids therefore requires an upfront preparative strategy that must be automatable if it is to keep pace with the desired sample throughput. Anderson et al. (2004) developed a targeted peptide quantification assay for plasma proteins using high affinity antibodies to enrich target tryptic peptides and their cognate internal peptide standards (Stable Isotope Standards and Capture by Anti-Peptide Antibodies, SISCAPA)[22]. This 'addition-only' magnetic bead-based workflow is amenable to automation by liquid handling robots[23], and its reproducibility and robustness has been demonstrated on 784 DBS samples using an LC-MS platform[24]. Thus, the SISCAPA immunoenrichment process increases the target peptides signal while effectively suppressing non-specific background to enable an LC-free approach required by AEMS.

Here, we show the application of two different SISCAPA assays on two very distinct biological matrices, shedding light on two aspects of

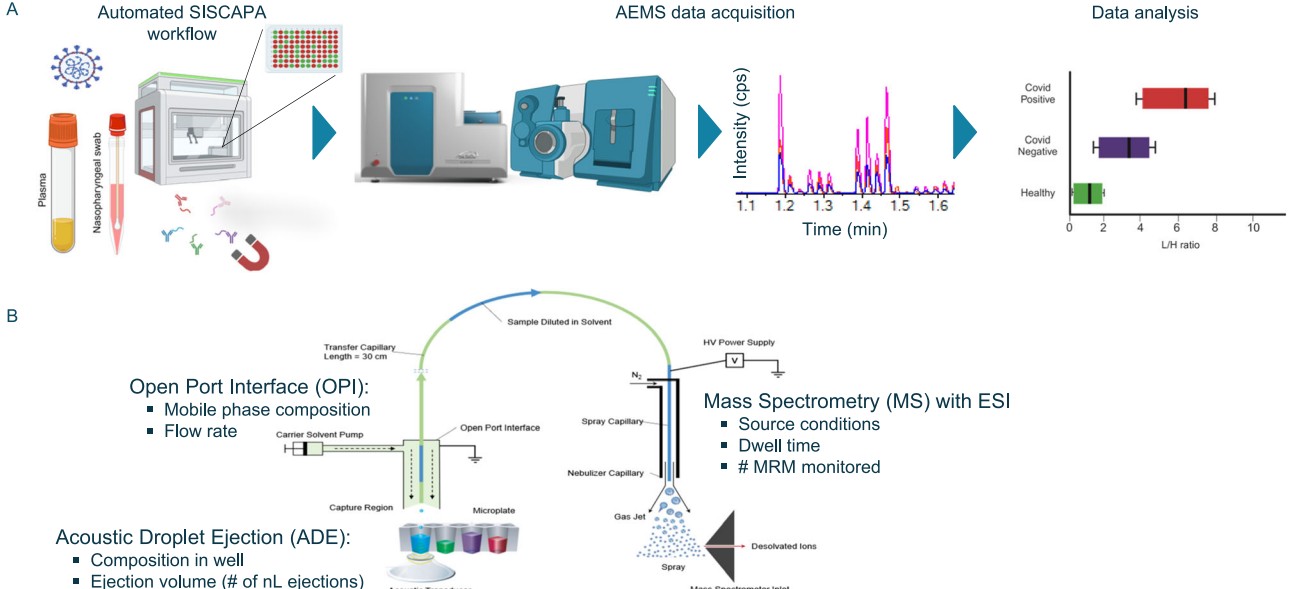

**Fig. 1 | High-throughput peptide quantification using peptide immunocapture on an Acoustic Ejection Mass Spectrometry (AEMS) platform. A** SISCAPA high affinity anti-peptide antibodies coupled to magnetic beads extensively purify peptide targets of interest from plasma and nasopharyngeal samples, which after acidic elution are ready for analysis by AEMS. In this study, the resulting data was used to monitor SARS-CoV-2 infection dynamics on two sample cohorts. Figure 1A, created with http://BioRender.com, released under a Creative Commons Attribution-NonCommercial-NoDerivs 4.0 International License. **B** In the first step of AEMS, ADE creates a standing wave in a sample well, and when the amplitude reaches a critical point, a single droplet is ejected. Sample volumes ranging from 2.5 nanoliters (nL) to hundreds of nL can be ejected with high reproducibility from a 384-well plate. This stream of droplets is captured in the Open Port interface (OPI) which is a pair of coaxial tubes, where carrier solvent flows through the outer tube and nebulizing gas draws the solvent through the inner tube to the electrospray ionization (ESI) source creating a solvent vortex. Acoustic ejection directs sample droplets into this vortex, facilitating rapid dilution and transfer to the mass spectrometer. Mass spectrometry at high acquisition rates is performed across the sharp ejection peaks for all samples from a 384-well plate, allowing a peptide to be quantified at a rate of 1–2 s per sample. Figure modified from Zhang et al.[17].

a SARS-CoV-2 infection, namely viral load determination and acute phase response to the infection. For the latter, a 10-plex SISCAPA assay, previously developed by Razavi et al. (2016), to quantify APR proteins from 267 plasma samples is employed (Supplementary Table 1), while a SISCAPA assay targeting the SARS-CoV-2 NCAP protein is adapted for detecting viral particles from 142 nasopharyngeal swab samples obtained from different individuals[9]. Both assays are optimized using an automated protocol tailored for the Beckman i7 automation workstation. The methods for peptide analysis are refined on the Echo MS system providing two distinct inherent advantages: absolute specificity and high-fidelity quantification, surpassing other antibody-based and colorimetric protein assays. A remarkable analysis rate of 1.5 s per peptide per sample is achieved with AEMS. Rigorous assessment of the specificity, precision and robustness of the optimized approach is conducted on distinct samples, illustrating the potential for protein biomarker measurements in cohorts potentially exceeding tens of thousands of samples prepared from various biofluids.

## Results

A substantial amount of optimization was performed (outlined in Supplementary Methods) to adapt the existing LC-MS assays to the AEMS platform for multiplexed protein quantification. Carrier solvent composition, flow rate, and sample ejection volume were optimized using a commercial standard peptide mixture (PepCalMix, SCIEX, Concord) in a simple protein digest (Beta-Galactosidase) to mimic anticipated sample complexities post-SISCAPA enrichment. Fine-tuning of the ejection volume was conducted, optimizing for peak shape and maintaining linearity between ejection volume and MRM peak area (Supplementary Fig. 2). Finally, the tolerance towards reagents used in the original (e.g. PBS, CHAPS) and modified SISCAPA protocol (ABC) was evaluated by investigating a range of concentrations on the ejection peaks (Supplementary Fig. 3). The narrow ejection peaks of AEMS necessitated careful optimization of the MS method. The chosen strategy involved 4 MRMs per method with a 10 ms dwell time, ensuring adequate data points across the ejection peak for precise quantification (Supplementary Fig. 4). Finally, these optimized parameters were used to determine the LLOQs for the PepCalMix peptides on two different AEMS systems, and good linearity

was observed for 19/20 peptides (average R2 of 0.997) with average LLOQs of 260 and 520 amol/μL, respectively (Supplementary Fig. 5 and Supplementary Table 6). A similar approach was applied for the ten peptides of the APR panel (Supplementary Table 7).

### Assessing platform reproducibility for peptide quantification

The reproducibility of an analytical system is key to its ability to characterize human biological variability. A mixture of the light and heavy peptides of the SARS-CoV-2 NCAP peptide AYNVTQAFGR (5 fmol/μL) in elution buffer was prepared and aliquoted across a 384-well plate (with 1 Marker Well). The peptide peak areas were recorded for the 383 sample wells with one ejection per well, then the plate was run 30 consecutive times acquiring 11490 sample ejections in total. This dataset comprises both the plate reproducibility and the long-term stability of the Echo MS system (Fig. 2A). The peak area reproducibility was found to be 4.6 and 6.4% CV for the light and heavy peptide, respectively, and the L/H peak area ratio for the peptide was maintained between 6.2–7.2% CV across the 11490 sample measurements (Fig. 2B). Data acquisition time for one plate was 10.5 min, so 30 plate runs required 5.25 h. This represents a sample acquisition rate of over 2000 samples per hour at very high reproducibility.

### Optimization of automated immunoenrichment sample preparation

The SISCAPA protocol has previously been optimized on various automation stations for LC-MS analysis[23,25,26]. Here, the protocol was adapted for the Biomek i7 automation station to make it compatible with AEMS analysis (Supplementary Fig. 1). To further improve the efficiency of bead washing and reduce the remaining salt concentrations, the buffer used during bead washing was switched from phosphate buffered saline (PBS) to ammonium bicarbonate (ABC), the plate type was changed from deep well U-bottom plates to deep well V-bottom plates to assist in the liquid removal, and new tips were used for each buffer removal step. Total time for the full sample preparation was ~6 h (including a 3-h digestion step at 37 °C), with final transfer of the 96 wells into a ready-to-analyze Echo qualified 384-well plate. The reproducibility of the total workflow, comprising the i7 sample preparation step, and the AEMS measurement was assessed by processing

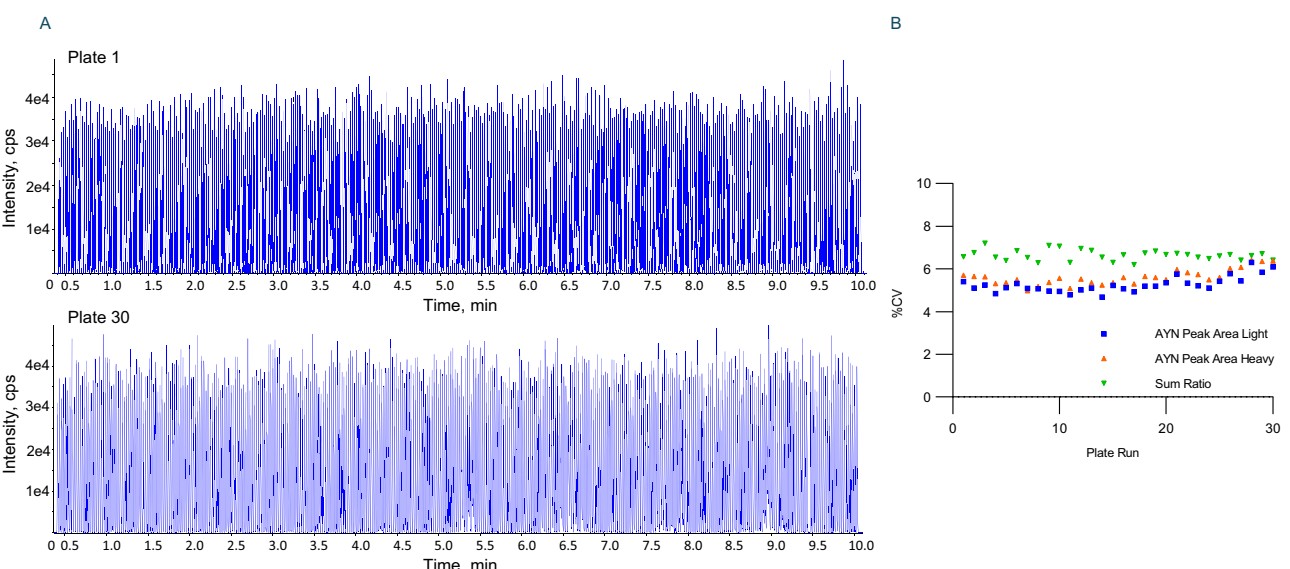

**Fig. 2 | Reproducibility of replicate AEMS peptide ejections over 5.25 h. A** The measurement of a peptide across 384 sample wells at the acquisition rate of 1.5 secs per well requires 10.5 min to acquire. **B** This measurement of the complete plate was repeated 30 times (acquiring 11490 sample ejections in total) and the peak area and peak area ratio percent coefficient of variation (%CV) were determined for each plate run. The %CV for the peak areas for the light and heavy peptides (Light AYNVTQAFGR (blue), heavy AYNVTQAFGR (orange)) were between 4.6 and 6.4% and the peak area ratios (summed L/H peak area ratio (green)) were between 6.2 and 7.2%.

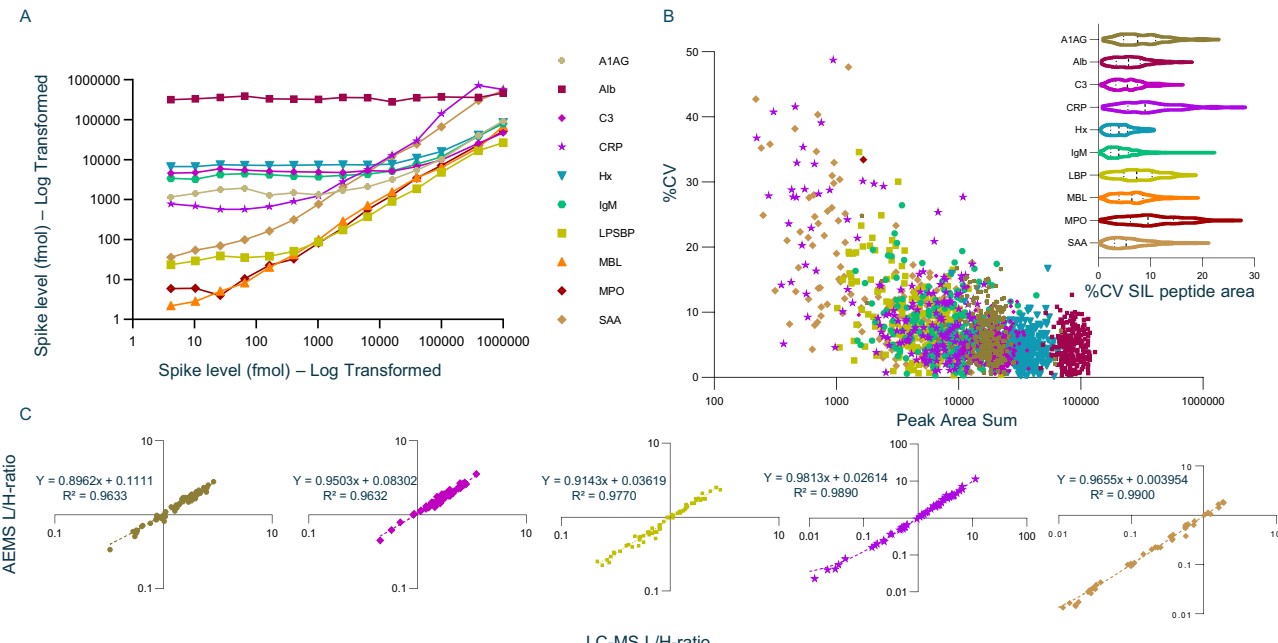

**Fig. 3 | Performance of AEMS for 10-plex APR assay. A** The dynamic range of the 10-plex APR was explored in pooled human plasma using a standard addition curve, spanning a range of almost >1000000, highlighting their varying abundance levels and potential implications for physiological processes. **B** Reproducibility of endogenous peptide areas for the acute phase response peptide areas from plasma captures ($n = 268$). As expected, the reproducibility observed for the endogenous peptides from triplicate technical measurements correlates with the observed peptide area or abundance, with the higher abundance peptides showing very good reproducibility (<10%) and the less abundant, lower area peptides having more variance. Data was subjected to the outlier rejection strategy and rejected data points were not plotted. Inset shows the reproducibility of acute phase response SIL peptide areas from plasma captures, in a violin plot. Data are represented by the median, first and third quartiles, and range. The reproducibility of the SIL peptide for each enriched sample measured in triplicate was found to be very good, with average %CV values across the 267 measured samples between 4.2% and 10.5%. **C** Correlation of measured AEMS L/H peptide ratios with LC-MS data ($n = 71$) for the most biologically relevant acute phase response proteins, namely A1AG, C3, LBP, CRP and SAA. The ratios measured by LC-MS were very similar to the ratios determined using the Echo MS system. After outlier rejection, the slopes for all proteins were very close to 1 and the $R^2$ values were 0.96 and higher.

16 wells of pooled healthy human plasma and using a pool and split strategy. The total workflow %CV for the light peptides from the 10 APR proteins ranged from 4.9–11.9% (Supplementary Table 5). The imprecision of upstream SISCAPA sample preparation ranged from 2.0–7.2%, encompassing both the imprecision arising from digestion variability and the liquid handling.

## Acute phase response proteins enriched from plasma samples—data quality

After optimization of the automated sample preparation protocol for immunoenrichment of the 10-plex APR peptides, a 14-point standard addition curve was created for the 10-plex to establish the endogenous levels of the ten analytes within pooled human plasma. The endogenous levels of the analytes, observed by the plateau in Fig. 3A, encompass a range spanning over six orders of magnitude. Next, a cohort of 225 plasma samples from confirmed COVID-19 positive and negative subjects, as well as 23 healthy plasma samples and one standard healthy plasma pooled sample (19 technical replicates) were processed. The samples were randomized across three 96-well plates for sample preparation, with one plate being processed per day on the Biomek i7 workstation. The %CV for the L/H-ratio sum for the 10 APR proteins ranged from 6.68–40.87% (Supplementary Fig. 6).

Datapoints for downstream biological measurements were deemed irreproducible and removed if the L/H peak area ratio between the two fragments monitored per peptide was greater than the average fragment ratio difference plus 2-sigma. A dilution series was additionally used to define the area observed at the LLOQ of each peptide, and values below this area threshold were also removed (Supplementary Fig. 7A). After application of this outlier rejection process, the reproducibility of the heavy peptide peak area (sum of

both fragments monitored) across triplicate measurements (Fig. 3B inset) was between 4.2% and 10.5% CV.

In Fig. 3B, the peak areas for the endogenous light peptides were plotted vs the triplicate %CVs. As expected, a strong correlation between the peak area and the reproducibility was observed. Six targeted peptides were easily detected by AEMS across all conditions, SAA and CRP were easily measured in unhealthy samples, which contrasted with healthy samples in which they were mostly undetectable[27,28]. The peptides from MPO and MBL protein were near or below the LLOQ in all samples. Supplementary Fig. 7B shows the proportion of datapoints per protein that were rejected across the entire dataset.

The utility of applying the SISCAPA workflow for protein quantification has been demonstrated using LC-MS analysis, MALDI analysis, and rapid trap-elute strategies[29–31]. Therefore, to benchmark AEMS, a subset of the samples were also run by LC-MRM on a SCIEX QTRAP 6500+ system using microflow chromatography. For eight peptides with quantifiable ratios, of which 5 are shown in Fig. 3C, a near-perfect linear correlation between LC-MS and AEMS was observed with average slopes of 1.022 and average $R^2$ values of 0.979. The linear correlation and regression between LC-MS and AEMS for the three other proteins (Alb, Hx and IgM) are shown in Supplementary Fig. 8.

The ejection volume and time between ejections was optimized based on the observed peptide signal and by the final plate, optimal ejection intervals were found to be 1.5 or 2 s, while the ejection volumes used were either 100 nL (Hx, Alb), 200 nL (C3, IgM, A1AG, LBP, MBL and MPO) or 300 nL (CRP and SAA). The time required to process all the samples in triplicate from the 96-well based sample preparation, using an ejection time of 1.5 s, was 1.6 h for a total of 4.8 h to analyze all three 96-well plates. Nevertheless, the robust reproducibility observed

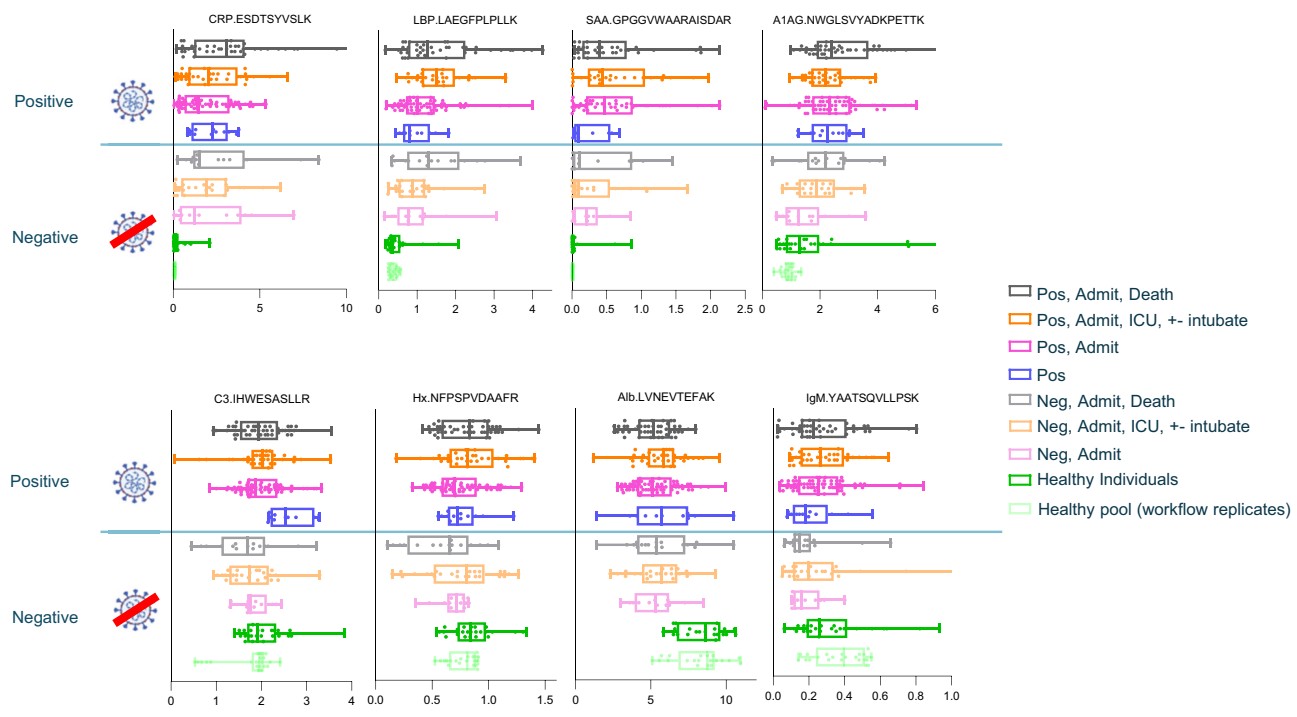

**Fig. 4 | Measured ratios of the eight quantified acute phase response protein in plasma cohort ($n$ = 268).** Samples were classified according to SARS-CoV-2 status (Positive/Negative) and disease severity (Supplementary table 8), then the light endogenous peptide ratio to the stable isotope labelled peptide was plotted (L/H ratio). Data are presented as following: minima, the smallest data point within 1.5 times the interquartile range (IQR) below the first quartile; maxima, the largest data point within 1.5 times the IQR above the third quartile; centre, the median of the dataset, representing the midpoint of the data; bounds of box, the lower and upper bounds of the box represent the first quartile and third quartile respectively, defining the IQR; whiskers, extending from the bounds of the box to the minimum and maximum values within 1.5 times the IQR from the first and third quartile respectively; and percentile, where Q1 represents the 25th percentile and Q3 represents the 75th percentile of the dataset. Four proteins were seen to increase in the disease samples (CRP, LBP, SAA, A1AG), and one protein (ALB) was found to decrease slightly in all disease samples. Figure 4, SARS-CoV-2 icon, created with http://BioRender.com, released under a Creative Commons Attribution-NonCommercial-NoDerivs 4.0 International License.

in the SIL peptide area (Fig. 3B, inset) serves as evidence that triplicate measurements are dispensable. This revelation presents a noteworthy prospect for a substantial reduction in analysis time, allowing for the completion of a single 96-well plate in just 32 min. In contrast, the run time per sample for the LC-MS assay was 5.5 mins to acquire all 10 peptides. For triplicate analysis, the total LC-MS run time for the cohort would be 73.5 h, meaning the Echo MS system workflow for analysing SISCAPA enriched samples is ~15x faster than microflow LC-MS, with identical peptide biomarker results and adequate signal.

**Acute phase response proteins and COVID-19 disease severity**
The cohort of plasma samples was collected from ill individuals admitted to either the medical floor or the ICU of the Cedars-Sinai Medical Center between March and May 2020. Samples were classified as positive if subjects had tested positive for SARS-CoV-2 within the first few days after admission, while the negative group consisted of symptomatic individuals whose RT-qPCR results were negative. Commercially purchased pooled healthy plasma and a cohort of 23 plasma samples collected from healthy individuals 5 years prior to the outbreak of the COVID-19 pandemic (2015) were both included. Samples were classified into nine distinct sub-groups by the Immunophenotyping Assessment in a COVID-19 Cohort (IMPACC) study and the WHO R&D Blueprint and COVID-19 classifications[32,33] (Supplementary Table 8).

The L/H peptide ratios were plotted according to their disease classification (Fig. 4). As expected, CRP was substantially higher in unhealthy samples (COVID-19 positive and negative) relative to healthy individuals. Interestingly, there was minimal difference in CRP levels between the COVID-19 negative and COVID-19 positive samples. LBP and SAA were also found to have a smaller but statistically significant increase in the unhealthy samples. A1AG gradually increased from healthy to severely ill SARS-CoV-2 samples. Other proteins (C3, Hx, and IgM) did not show a significant change in this cohort, although the C3 protein in the SARS-CoV-2 infected, non-admitted class of samples seemed to show a spike which requires confirmation in future studies on larger populations. Finally, Alb showed small decreases in abundance among the unhealthy samples in the cohort, also consistent with its role as a negative acute phase reactant.

Our results were consistent with Messner et al. (2020)[4], who found that CRP, LBP and SAA1:SAA2 were increased in COVID-19 positive samples, with some separation between the COVID positive and severe disease samples. This group similarly reported Alb down-regulated with COVID-19 disease. Other studies also found SAA1:SAA2 and CRP upregulated in COVID-19 disease[3,34]. Razavi et al. (2016) followed 22 proteins in 14 individuals over time from self-collected dried blood spots[35], and observed large elevations in SAA, CRP, and LBP levels upon symptomatic undefined (presumably viral) infection, along with a small persistent drop in albumin levels consistent with an acute phase response. A similar pattern of expression was observed in our cohort, with elevated levels of SAA ($p < 0.0001^*$), CRP ($p < 0.0001^*$), and LBP ($p < 0.0001^*$) alongside a decrease in ALB ($p < 0.0001^*$) in all unhealthy individuals presenting ill at the hospital, regardless of

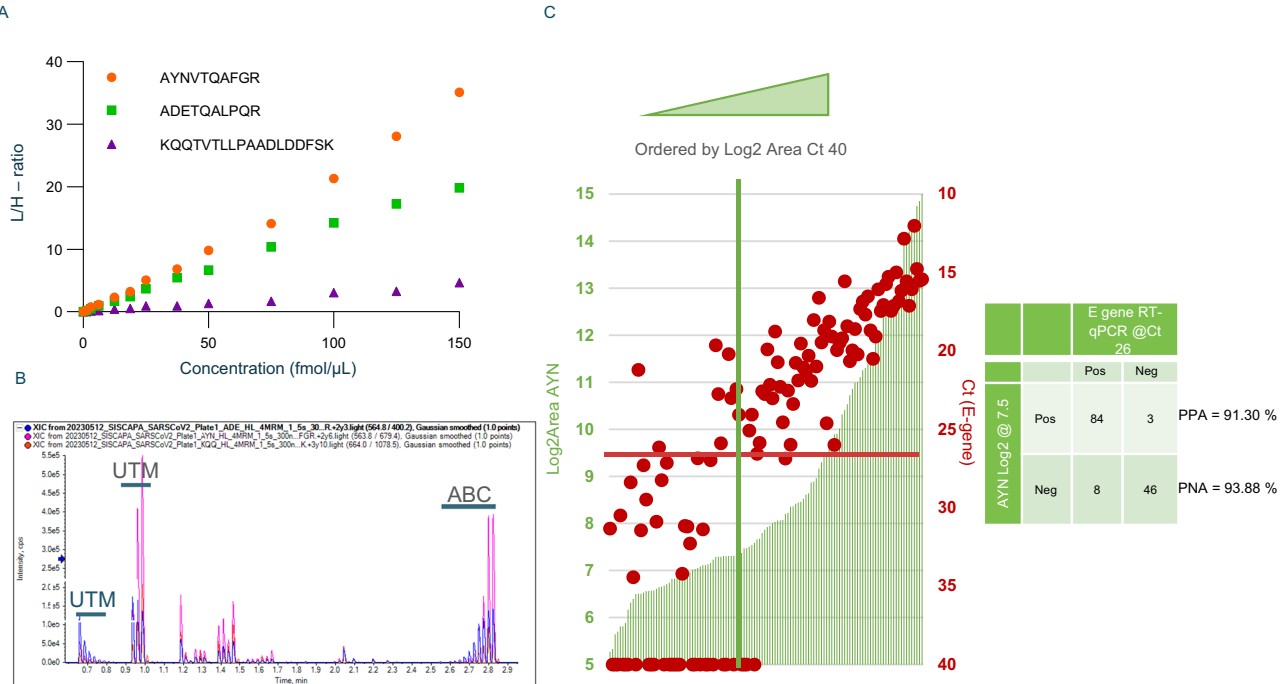

**Fig. 5 | Evaluation of the SARS-CoV-2 NCAP assay with AEMS. A** Linearity of the average summed Light to Heavy (L/H) ratio with the spiked concentration of NCAP protein in Universal Transport Medium (UTM). **B** Overlay of the XIC of the three target peptides across samples with a prior dilution series in UTM and ending with a dilution series in ammonium bicarbonate (ABC). Each peak represents one patient sample. **C** Secondary axis plots of the raw measurements of E-gene Cycle threshold (Ct) (red dots) and AYNVTQAFGR logarithmically transformed MS Peak Area (Log2Area) (green bars) for results sorted from low to high Log2Area. A strong linear correlation illustrates the level of agreement between RT-qPCR and AEMS, with Log2Area flattening at 7.5 (green line), i.e. beyond Ct > 26 (red line). A high percent positive (PPA = TP/(TP + FN)) and negative agreement (PNA = TN/(TN + FP)) between RT-qPCR (Ct) and MS (AYN Log2Area) is achieved, especially below Ct 26.

COVID-19 status (Fig. 4). A similar correlation between CRP and SAA was observed, as well as between CRP and LBP and between SAA and LBP proteins. However, since the ratios among CRP, SAA, and LBP change dramatically during the time course of an infection (Supplementary Fig. 9), the significance of this finding remains to be confirmed[35]. As this is only a discovery cohort of 225 samples, other comparisons did not directly result in statistically significant differences. Moreover, it's worth highlighting that with the increased speed of data acquisition facilitated by AEMS, such comparative analyses comprising thousands of samples can now be performed more readily, potentially enhancing the precision of our findings in future studies.

## Quantification of SARS-CoV-2 NCAP peptide enriched from nasopharyngeal swabs

Recently, the capability of the SISCAPA assay in combination with LC-MS to detect the NCAP protein from up to 500 nasopharyngeal swabs per day up to the limit of infectiousness was shown, corresponding to an estimated RT-qPCR Ct-value of 32-33[8,9]. Considering the overwhelming number of SARS-CoV-2 tests performed worldwide during the pandemic, the prospect of applying AEMS for measuring tens of thousands of samples a day on a single platform could greatly increase society's pandemic preparedness. A small cohort of 145 nasopharyngeal swab samples was processed in two 96-well plates using the semi-automated sample preparation protocol described above, including an additional SPE step for further removal of background matrix (see Supplementary Fig. 10 for Ct distribution).

The sensitivity of the AEMS method was tested on the three SARS-CoV-2 NCAP peptides (AYNVTQAFGR, ADETQALPQR, and KQQTVTLLPAADLDDFSK) from a dilution series of recombinant SARS2_NCAP protein (0–150 fmol/μL) in the commonly used UTM medium. With the Echo MS system, an LLOQ in UTM of 0.195 fmol/μL

was achieved for both the AYNVTQAFGR and ADETQALPQR peptides (Fig. 5A), while KQQTVTLLPAADLDDFSK demonstrated a higher LLOQ of 3.12 fmol/μL. The respective peak area values were used for outlier removal (Supplementary Fig. 11). In a previous study using LC-MS, excellent linearity down to 4 amol/μL was observed, corresponding to 144 mol on column[9]. This LC-MS method applies 10 μL on column, whereas the ejection volume with AEMS was 300 nL, resulting in a 33.3-fold higher loading in LC-MS. It is this sample loading difference that explains the bulk of difference in assay sensitivity between LC-MS and AEMS. Very good reproducibility of the triplicate measurements of the SIL peptides was observed (Supplementary Fig. 12), with average %CV values of 5% to 6.5%.

A selection of samples was run by LC-MS, and again very good correlation was observed between the peptide ratios measured by AEMS and LC-MS, (Supplementary Fig. 13) with average slopes of 0.983 and an $R^2$ of 0.943. Notably, while AEMS is less sensitive compared to LC-MS and therefore starts losing accuracy at the lowest peptide concentration, the variation in the correlation plot for the KQQTVTLLPAADLDDFSK peptide derives from the LC-MS runs, which suffer from column carry-over for this peptide, as described earlier (9). Interestingly, as there is no chromatography with AEMS, this peptide behaved very well in the Echo MS system. Next, a total of 142 nasopharyngeal swab samples were screened using the AEMS assay for SARS-CoV-2 NCAP peptides. An overlay of the Extracted Ion Chromatograms (XIC) for each peptide run (with dilution series before and after the sample batch) show the intuitive nature of the data (Fig. 5B), with each visible peak indicating an infection measured in 1.5 s.

A binary comparison between RT-qPCR for the E-gene (red dots, right axis) and Log2Area of the AYNVTQAFGR peptide (green bars, left axis) is shown in Fig. 5C. The cohort comprised 113 qPCR positive and 29 RT-qPCR negative samples, with a Ct value >40 being considered

negative. In accordance to the Clinical and Laboratory Standards Institute (CLSI) user protocol for evaluation of qualitative test performance (EP 12-A2), percent positive and negative agreement (PPA and PNA) were calculated. The PNA/PPA matrix in Fig. 5C depicts these numbers when a summed intensity of AYNVTQAFGR of 7.5 is used. However, there are several concerns with this representation as previously described by Van Puyvelde et al., therefore results were sorted from low to high virus measurement, i.e., from low to high Log2Area, which inversely correlates to the Ct value[9]. A clear correlation between both tests was found, as described earlier for LC-MS measurement. When a Log2Area of 7.5 for AYNVTQAFGR was applied as a cutoff (see dilution series in Fig. 5A), eight samples with Ct < 26 are wrongly classified as negative (PPA = 91.30%). Whether these outliers have an analytical cause (in the RT-qPCR or the AEMS assay) or whether this is a case of residual RNA[36], cannot be addressed at this point. Inversely, at this threshold, 3 samples with Ct 40 were classified as positive by AEMS but as negative based on the RT-qPCR Ct value (PNA = 93.88%). Overall, within the current small sample cohort, Ct 26 seems a good first estimate for the NCAP detection limit by AEMS in nasopharyngeal swabs. This aligns well to the published LC-MS method, where a threshold of Ct 30-32 in UTM could be attained, but now with 33-fold lower sample loading on AEMS ($2^5 = 32$ and thus 5 Ct values lower). However, AEMS provides a potential rate of 2400 samples per hour, as opposed to 30 per hour using LC-MS[9].

## Discussion

The rapidly evolving landscape of epidemiological, population sciences, and clinical research demands advanced analytical techniques that can efficiently handle large numbers of samples while maintaining high-throughput, precision, and sensitivity. Here, the promise of Acoustic Ejection Mass Spectrometry (AEMS) for the precise and accurate peptide quantification at ultra-high sample throughputs has been demonstrated, theoretically attaining 50,000 peptide measurements per day per AEMS platform for single peptide viral detection in nasopharyngeal swabs, or over 5000 plasma samples (or 1700 samples in triplicate) per day, targeting 10 different APR protein biomarkers.

Here, the sample simplification power of SISCAPA immunocapture to enrich target peptides from matrices has been combined with direct MS analysis using AEMS. Using smaller COVID-19 cohorts, the focus was to fully characterize the data quality obtained from this high-throughput workflow. The demonstrated workflow reproducibility, from sample preparation of plasma to MS analysis of peptide replicates was very encouraging with total workflow and AEMS %CVs between 4.9 and 11.9%, and an excellent correlation with LC-MS ($R^2$ values ≥ 0.96). Similar responses were observed, with CRP, SAA, LBP, and A1AG increasing with infection, which compared favorably with previous COVID-19 publications. The nasopharyngeal swabs were also screened for SARS-CoV-2 by RT-qPCR and good correlation was observed down to Ct = 26.

Out of the ten APR peptides targeted from 10 μL of plasma, only eight were successfully quantified. By utilizing larger volumes of plasma and scaling the antibody capture accordingly, it is possible to target lower abundance proteins, as demonstrated by the SISCAPA assay for Thyroglobulin[37,38], as well as by the stable signal that was measured for the Cov²MS assay on dilution series for nasopharyngeal swabs[9]. The antibody-bead and heavy peptide amounts did need to be adjusted to improve the performance of the CRP and SAA peptides in the final assay, suggesting that custom-designed antipeptide antibody mixtures could be developed for AEMS, achieving an optimal balance between assay performance and assay cost.

With the MS throughput now possible by AEMS, sample preparation becomes the bottleneck, underscoring the imperative for automation[39]. The Biomek i7 automated station configuration used here can currently process > 300 samples per 24 h, requiring several liquid handling robots to continuously feed one AEMS platform using the protocol described here[25]. While it was not tested here, it is feasible to decrease trypsin incubation to a 30 min interval, or to perform this step offline to reciprocally increase the number of 96-well plates that can be processed daily by a single operator[12]. Using automation stations equipped with multiple shakers and incubators, processing two 96 well plates in 4.5 h, or ten 96-well plates per day is possible[25]. Therefore, with more advanced automation stations, fully automated sample preparation is achievable, and coupling the Echo MS system with automatic plate loading systems can enable continuous 24/7 operation[40].

There have been a wide range of immune-enrichment assays developed for peptides, PTMs, and proteins that have been applied to various biological matrices that could be potentially coupled with AEMS analysis (such as serum[38], tissue[41], and urine[42]). One area of particular interest for epidemiology studies is using dried blood spots (DBS), collected by individuals at home longitudinally, and subsequently processed by the SISCAPA workflow for MS analysis. In one study, 16 subjects collected DBS longitudinally to generate 1662 samples[24,35].

However, this notable increase in throughput comes with an analytical cost, i.e. a lower sensitivity. While the manuscript currently lacks a direct LLOQ comparison with a comparable high-flow LC-MS approach, using (i) a dilution series of recombinant protein in UTM and (ii) the same patient samples and their known Ct values measured earlier[9] as a proxy, a roughly 40-fold reduction in sensitivity was found, which is partially due to differential sample loading amounts. Considering the sensitivity of the latest generation of MS instruments, the sensitivity of AEMS is comparable to what LC-MS assays could attain less than ten years ago. In turn, this implies that employability of the assay needs to be thoroughly assessed for each envisioned application. For example, a winterplex panel that simultaneously quantifies two peptides for each of the following respiratory viruses: Influenza (A and B); RSV; and SARS-CoV-2 is currently under development. Targeting these eight peptides would still allow the measurement of over 6500 samples on one platform in a 24-h workday. In an early stage of a new pandemic, this platform could greatly facilitate the study of viral spread within the population.

In conclusion, the combination of automated immunoenrichment sample preparation with AEMS for protein quantification opens exciting possibilities for accelerated biomarker validation, ultimately advancing our understanding of diseases and enabling timely public health interventions. Further refinements and scaling of this automated and integrated workflow, coupled with expanded studies involving diverse cohorts and longitudinal sample collection, will be crucial for realizing the full potential of this approach and creating a paradigm shift in biomarker research and clinical applications, ushering in a new era of rapid and precise proteomic analysis with far-reaching implications for personalized medicine and population health.

## Methods

The research conducted in this study adheres rigorously to all pertinent regulations, including the criteria outlined by the Declaration of Helsinki, concerning the involvement of human study participants. The AEMS and LC-MS data generated in this study have been deposited to the ProteomeXchange Consortium via the Panorama Public partner repository with dataset identifier PXD046249[43].

### Reagents and materials

The standard samples used for method optimization (PepCalMix and Beta-galactosidase digest) were obtained from SCIEX (Framingham, USA). Echo qualified 384-well poly-propylene microplates were obtained from Beckman Coulter Life Sciences (San Jose, USA).

Unlabeled (light) and stable isotope labeled (SIL or heavy) peptides for the different peptide targets were obtained from Biosynth (Staad, Switzerland) and SISCAPA Assay Technologies (Victoria, Canada). Upon receipt from the vendor, the lyophilized peptides were solubilized in 30% acetonitrile (ACN)/0.1% formic acid (FA) and stored frozen at −80 °C. Dynamic range optimized monoclonal anti-peptide antibody mixtures coupled to magnetic Dynabeads Protein G (Thermo Fischer Scientific, MA, USA) were sourced from SISCAPA Assay Technologies (Victoria, Canada) and stored at 4−8 °C. Recombinant NCAP of SARS-CoV-2 (2019-nCoV, P/N: 40588-V08B) was produced in insect cells with a baculovirus expression system (Sino Biological, Beijing, China). Urea, Trizma pre-set crystals, Tosyllysine Chloromethyl Ketone (TLCK) and 3-[(3-Cholamidopropyl)dimethylammonio]-1-propanesulfonate (CHAPS) were sourced from Sigma-Aldrich (St. Louis, USA) while Tris (2-carboxyethyl)phospine (TCEP Bond-Breaker neutral pH) and Iodoacetamide (IAA) were obtained from Thermo Fischer Scientific.

## Samples

Remnant plasma samples from the Cedars-Sinai Medical Center (CSMC) Biobank Resource, from whole blood collected in anticoagulant-treated (EDTA) tubes, were obtained from a single-center observational study of adult subjects admitted to either the medical floor or the intensive care unit (ICU) of CSMC between March 23 and May 10, 2020, and who were experiencing symptoms related to Covid-19. The samples selected for this study were collected within the first 1−2 days from admittance. Ethical approval was obtained from the Institutional Review Board on Research Involving Human Subjects (CSMC IRB#: Pro00036514; Cedars-Sinai Remnant Biobank). In addition, gender pooled (100 females and 100 males) EDTA human plasma from healthy individuals (abbreviated as JVE Plasma) was purchased from BioIVT (Westbury NY, USA).

Residual Covid-19 nasopharyngeal samples were obtained from the AZ Delta Hospital, Roeselare, Belgium, with approval of the University Hospital Ghent ethics committee (BC-09263). Both studies were performed in accordance with the Helsinki declaration.

## Sample preparation

**10-protein multiplex acute phase response (APR) assay.** The sample preparation workflow described by Razavi et al. was altered to make it compatible with the Biomek i7 automated workstation (Beckman-Coulter, CA, USA) preconfigured for protein denaturation, cysteine reduction, alkylation, trypsin digestion and sample desalting and the AEMS analysis workflow[23,44,45]. Briefly, 17 µL of a 'denaturation mix' (9 M urea, 0.05 M TCEP and 0.2 M Trizma) was aliquoted into each V-shaped well of a 96 well plate (Prod. No P-96-450V-C-S; Axygen). The mixture was then incubated overnight in a dry incubator at 37 °C, allowing the mixture to dry in the wells. The next morning, 10 µL of human plasma (BioIVT, Westbury NY, USA or from the CSMS cohort) was added to each well followed by a 30-min incubation on a ThermoMixer F (Eppendorf, Hamburg, Germany) at room temperature with vigorous mixing at 600 rpm. Although sample addition is automatable, here we manually transferred the plasma to prevent sample loss from aliquoting. All subsequent steps were performed with a Biomek i7. Cysteine alkylation was performed by adding 10 µL of a 0.05 M IAA solution, followed by an incubation of 10 min at room temperature in the dark (dark lid covering the plate). Following dilution with 115 µL of 0.2 M Trizma buffer to reach a final urea concentration of 1 M, 10 µL of 7.3 mg/mL trypsin (Worthington) in 10 mM HCl was added. The 96-well plate was then incubated for 3 h at 37 °C on a shaking incubator (Inheco, Germany) that is integrated onto the Biomek i7 deck. Tryptic cleavage was stopped by adding 10 µL of 0.22 mg/mL TLCK in 10 mM HCl followed by a 5 min incubation at room temperature. Before the trypsin digested plasma samples were subjected to SISCAPA peptide

enrichment, 10 µL of a 10-plex SIL mixture (Supplementary Table 1) was added. The dosing of heavy peptides was optimized during method development to ensure good signal being observed for both disease and healthy controls. The deck layout for the i7 workstation is shown in Supplementary Fig. 1.

A total of 249 clinical samples were subjected to the automated protocol, comprising 225 samples from hospital-admitted individuals and 24 from healthy subjects. Additionally, 19 technical replicates of a gender-pooled plasma sample were included in the analysis.

**SARS-CoV-2 peptide detection.** Sample preparation was performed as outlined by Van Puyvelde et al., wherein proteins in 180 µL of undiluted Bioer Universal Transport Medium (UTM) underwent protein precipitation by the addition of 1260 µL of ice-cold acetone[9]. After centrifugation (16,000 g) in a cold environment (0 °C) for 10 min, the supernatant was discarded and the residual protein pellet was resuspended in 140 µL of a trypsin-Lys C (Promega, Madison, WI, USA) digestion buffer (0.007 µg/µL in 100 mM NH₄HCO₃). After a 30-min incubation at 37 °C, trypsin activity was inhibited by adding 20 µL of a 0.22 mg/mL TLCK solution in 10 mM HCl. Finally, 5 µL of a SIL mixture (55 fmol/µL) of the three NCAP peptide targets (ADETQALPQR, AYNVTQAFGR and KQQTVTLLPAADLDDFSK) was added before peptide enrichment[9]. A total of 145 nasopharyngeal patient samples was prepared using this protocol, alongside a dilution curve comprising 12 samples.

## Peptide enrichment

Prior to the addition of the antibody-coupled magnetic bead immunoadsorbents, a step was included to fully resuspend the beads. To perform the SARS-CoV-2 peptide detection assay, magnetic beads coupled with monoclonal antibodies for three specific peptides for the NCAP protein, namely AYNVTQAFGR (Lot. #H03242101-1H2-AYN), ADETQALPQR (Lot. #H03192101-ADE), and QQTVTLLPAADLDDFSK (Lot. #H03102101-1G1-QQT) were combined in equal volumes (conc. of individual peptide antibodies in mixture: 0.0333 µg/µL), after receiving them from SISCAPA Assay Technologies. Next, 30 µL of this mixture was added to the nasopharyngeal swab samples after trypsin digestion. For the APR panel, 20 µL of the magnetic bead mixture coupled with monoclonal antibodies for the 10 target peptides, acquired from SISCAPA Assay Technologies (Supplementary. Table 1), was added to the digested plasma samples. The APR antibodies were: Hemopexin (Lot. #H10122103), Albumin (Lot. #H10122101), Immunoglobulin M (Lot. #H10192102), C-reactive protein (Lot. #H10192104), Alpha-1-Acid Glycoprotein (Lot. #H10192101), Serum Amyloid A (Lot. #H07251403), LPS Binding Protein (Lot. #06161504), Mannose Binding Lectin (Lot. #H04242009), Complement 3 (Lot. #H10122104), Myeloperoxidase (Lot. #10122102).

After a 1 h bead incubation step while shaking at 1000 rpm, washing and elution was performed as previously described, however a different wash buffer was used (10 mM ammonium bicarbonate (ABC), 5% MeOH, 0.00025% CHAPS) and one additional bead washing step, totaling four bead washing steps, was incorporated prior to elution[23]. For the nasopharyngeal swabs, an additional Solid-Phase Extraction (SPE) step was included after elution to further clean up the samples, using the HLB µElution kit (Waters, Milford, USA). Prior to AEMS analysis, the purified peptides in 50 µL of elution buffer (0.5% formic acid and 0.00025% CHAPS) were transferred to an Echo qualified 384-well plate, spun at 1000 g for 1 min to achieve a uniform fluid meniscus.

## AEMS analysis

All samples were analyzed in triplicate at 1−3 s per sample from a 384-well plate using MRM analysis on an Echo MS system with the SCIEX Triple Quad 6500 + mass spectrometer (SCIEX), operating under

SCIEX OS software (v.3.0.0.3339). AE conditions were set as following: a carrier solvent composition of 80% acetonitrile/200 nM medronic acid, a carrier solvent flow rate of 500 μL/min, and a droplet count of 100–300 nL total ejection volume. When measuring all ten APR peptides once with a droplet count of 300 nL, the analysis requires 3 μL, equivalent to 6% of the total sample volume. Considering that a minimum volume of 20 μL is necessary in each well of a 384-well plate for efficient sample ejection, this ensures 30 μL or up to 100 ejections for analysis.

The source conditions were set as following: Gas1 = 90, Gas2 = 70, CurtainGas = 25, SourceTemp = 400 °C, IonSpray voltage = 5000 V. Two MRM transitions (dwell time of 10 ms each) were monitored per peptide, both for the heavy and light peptides (Supplementary Table 2 and 3) plus an extra MRM to monitor CHAPS (dwell time 3 ms) in the elution buffer which served as the Marker well, required for successful file splitting[46].

## LC-MS analysis
An Eksigent NanoLC 425 System (SCIEX, CA) plumbed for capillary flow chromatography (10 μL/min) was used and operated in direct injection mode with the SCIEX Triple Quad 6500+ mass spectrometer. The column used was a 50 × 0.3 mm Kinetex XB-C18 Column (2.6 μm, 100 Å) and the temperature was controlled at 30 °C. The gradient used is outlined in Supplementary Table 4 for a total sample analysis time of 8.5 mins. For the inflammation panel samples ($n = 71$), 1 μL of the final sample was injected (3.3-fold higher sample loading compared with AEMS). For the SARS-CoV-2 sample ($n = 63$), the final sample was diluted ¼ with Mobile phase A to reduce the organic fraction (from 40% to 5% ACN) and 2 μL was injected on the LC-MS (1.66-fold higher sample loading compared with AEMS). The same two MRM transitions that were used for Echo MS analysis were used for LC-MS analysis, run using the Scheduled MRM algorithm (Supplementary Table 2). The source conditions were set as follows: Gas1 = 20, Gas2 = 20, CurtainGas = 35, SourceTemp = 100 °C, IonSpray voltage = 5000 V.

## Data processing
The MS signal from ejections of individual wells from the entire plate were recorded in a single raw data file then split post-acquisition into individual samples within the file, using the acoustic ejection log[46]. All samples were then processed using the Analytics module in SCIEX OS software (v3.0.0.3339). For the samples acquired with the Echo MS system, the integration algorithm used was Summation, with the Gaussian smoothing width set to 1 and the Noise % for baseline set to 0. For the LC-MS data, the MQ4 integration algorithm was used, again with a 1-point Gaussian smooth and the Noise % for baseline set to 0. As two MRMs were monitored per peptide (Supplementary Table 2 and 3), these were summed together to obtain a single summed peak area for each of the light and heavy peptides. In order to ensure high quality quantification in the dataset, an outlier rejection strategy was developed and applied to the data. Here the average difference observed between the L/H ratio for each of the two fragment ions monitored per sample was determined, then the average and standard deviation of those fragment ratio differences was computed across the sample preparation plate. The average fragment ratio difference + 2 standard deviations (sigma is standard deviation) was applied as a maximal allowed ion ratio difference to remove outliers from the dataset[47]. A minimum peak area threshold was also applied to the summed peak areas for both the heavy and light peptides, based on a dilution series of the light peptide targets in elution buffer for the APR peptides and UTM matrix for the SARS-CoV-2 peptides. The summed peak areas at the lower limit of quantification (LLOQ) were calculated, and then the minimum peak area threshold was set by rounding up to the next 500 value. All the data was imported in Skyline Daily (v.23.0.9.187), the freely available MS data analysis software, to allow data visualization for anyone without access to SCIEX OS

software. Visualization of the data was achieved with Microsoft Excel 365 (v2311) and GraphPad Prism (v.10.0.3).

## Reporting summary
Further information on research design is available in the Nature Portfolio Reporting Summary linked to this article.

## Data availability
The mass spectrometry proteomics data (AEMS and LC-MS) generated in this study have been deposited to the ProteomeXchange Consortium via the Panorama Public[48] partner repository with dataset identifier PXD046249[43]. Source data are provided with this paper.

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

## Acknowledgements

Funding from NIH 1U54CA260591-01 (JVE), NIH 1 U01 NS115658-01 (JVE), Erika Glazer Endowed Chair in Women's Heart Health (JVE), European Unions' Horizon Research and Innovation Program (No. 101073924) and Research Foundation Flanders (FWO) [1278023 N and V422522N] (B.V.P.). We are grateful to SISCAPA Assay Technologies Inc. for the support (incl. SIL peptides) received during this project. We would like to acknowledge the work of Cedars-Sinai Biobank and those involved in the collection and processing of samples of those hospitalized for SARS-CoV-2 infections.

## Author contributions

B.V.P., C.L.H., M.Z., Y.W., E.H., Q.F., M.D. and J.V.E. contributed to conceptualization; M.P, M.R., and L.A. provided support regarding the SISCAPA protocol; S.S. provided support with the Biomek i7 automation. B.V.P., C.L.H., M.Z., E.H., and Y.W. performed Echo MS experiments; C.L.H. performed LC-MS experiments; K.R. helped with resources and editing; G.M. provided SARS-CoV-2 patient samples; D.D. helped with funding acquisition; C.L.H., B.V.P, Q.F., M.D. and J.V.E contributed to writing (original draft) and all authors have proofread the manuscript.

## Competing interests

The authors declare the following competing interest(s): C.L.H. was an employee of SCIEX. S.S. is an employee of Beckman Coulter Life Sciences. M.P., M.R. and L.A. are employees of SISCAPA Assay Technologies. The remaining Authors declare no competing interest.
