## [Peer Review File · Nature Communications]

Reviewers' Comments:

Reviewer #1:

Remarks to the Author:

This manuscript presents a new method utilizing AEMS for the quantification of acute phase response protein markers and SARS-CoV-2 NCAP peptides in automated assays. The subject matter is relevant to the readership of Nature Communications, Nature Portfolio, and I endorse its publication pending the addressing of the following comments.

The article effectively optimizes the AEMS methods, addressing factors such as the repetitive readout of a particular plate for acute phase response proteins, requiring the plate to be read 10 times to analyze the entire set of 10 proteins, one protein at a time. Additionally, it optimizes parameters such as mobile phase composition, flow rate, sample ejection volume, dwell time, etc. However, a more detailed comparison of the LLOQ between AEMS and LC-MSMS is warranted. Firstly, the LLOQ of LC-MSMS is currently absent, and it was performed on the Sciex TQD system. Secondly, the previous result is not accurately presented in the paper. On Page 10, it states, "In a previous study using LC-MS, excellent linearity down to 4 amol/ μ L was observed, corresponding to 72 amol on column (9)." However, in line 99 of the previous publication (https://pubs.acs.org/doi/suppl/10.1021/acs.analchem.2c01610/suppl_file/ac2c01610_si_001.pdf), it was mentioned that 2 amol/ μ L corresponded to 72 amol on the column. Thirdly, the comparison of ejection amounts between AEMS and LC-MSMS should be corrected. On Page 10, it states, "This LC-MS method applies 10 μ L on column, whereas the ejection volume with AEMS was 300 nL, resulting in a 33.3-fold higher loading in LC-MS." However, the previous reference reports a 10 μ L injection with a Waters Xevo TQ-XS system (References #9), while the injection volume of the current Sciex LCMS system is 1 μ L for the inflammation panel samples and 2 μ L for the SARS-CoV-2 sample (Page 14). A thorough comparison of the Lower Limit of Quantification (LLOQ) between AEMS and LCMS is essential to assess whether AEMS can attain a sensitivity level suitable for the assays, particularly when using Mass Spectrometry systems from the same vendor, as opposed to comparisons across different vendors. In this context, there is a 3.3-6.6-fold higher loading in Sciex LCMS. Given this disparity, how does the LLOQ of LCMS compare to that of AEMS?

The L/H ratios of acute phase response proteins within COVID-19 disease cohorts serve as severity biomarkers to analyze the positive/negative status of COVID-19. Regarding the discussion of Figure 4 on Page 9, has statistical analysis been conducted for the increase in CRP, LBP, SAA1:SAA2, and the decrease in Alb in COVID-19 positive samples? For a thorough analysis, it is advisable to conduct a paired t-test in both positive and negative sub-cohorts (e.g., COVID Positive, Admitted, death vs. COVID Negative, Admitted, death, and other relevant sub-cohorts).

Figure 5 illustrates the comparison of samples analyzed by AEMS and RT-PCR. However, the accuracy evaluation of the SARS-CoV-2 NSAP assay with AEMS is not clearly elucidated for the 142 nasopharyngeal swab samples when using RT-PCR results as references. I am interested in understanding the percent positive agreement (PPA) and percent negative agreement (PNA), especially when $Ct < 25$ and $\text{Log}_2\text{Area AYN} > 7.3$ are considered positive. Previous reports (References #8 and #9) demonstrate consistent results between RT-PCR and LCMS, both adhering to the CLSI EP 12-A2 User Protocol for Evaluation of Qualitative Test Performance guidance. How does the current LCMS data compare to RT-PCR? Through these comparisons, we can gain insights into the predictable accuracy for COVID-19 among AEMS, LCMS, and RT-PCR.

Regarding the Solid-Phase Extraction (SPE) clean-up for peptide enrichment, the elution of the peptide involved $2 \times 10 \mu\text{L}$ of 40% acetonitrile in 0.1% formic acid, followed by dilution with 35 μL of a 0.5% formic acid solution in H₂O. Has there been an assessment through recovery experiments to confirm the accuracy and reproducibility of this step? Typically, after eluting the analyte from the SPE column, the common practice is to evaporate the solution and reconstitute it with a mobile phase solution or another buffer to maintain a consistent volume in the final samples. This step is essential because elution volumes often exhibit inconsistency in SPE,

regardless of whether a positive pressure or vacuum procedure is utilized. I understand that resolving the peptides by drying down and reconstituting may introduce additional challenges.

Furthermore, there is a need for improvement in the legend and axis units of Supplementary Figure 3. It is challenging to determine the accuracy of CHAPS and 1xPBS as the maximum tolerated concentration is presented in a different unit on the axis, which I presume to be mM. In the case of TRIZMA, the maximum tolerated concentration appears to be around 0.0006 mM, assuming a similar interpretation as the other two additives, NaCl and AmBic.

Reviewer #2:

Remarks to the Author:

This manuscript describes an Echo-MS method for high-throughput peptide quantification after sample enrichment using immunocapture from digested biological matrices. This is the first report of using Echo-MS technology for peptide analysis, and the authors have demonstrated excellent quality and reproducibility of the methodology. The manuscript could be further improved by addressing the following comments.

1. With a limit of 4 MRM channels per Echo-MS method, it seemed that each sample had to be analyzed multiple times, once for each H/L peptide pair. Does that mean the for the 10-plex SISCAPA extracted samples, each sample was analyzed 30 times (3 technical replicates x 10 peptide pairs)? The need for multiple Echo-MS analyses on the same sample was mentioned in the manuscript, but I think it would need to be clarified in the throughput discussion (page 8, line 215).

2. With a throughput of ~5 hrs/96-well plate on Echo-MS and 1 day/96-well for sample preparation, there's an obvious mismatch and the bottleneck was shifted to the automated the sample preparation step. I think it would be helpful to add some discussion on how to further improve the speed of digestion and immunocapture (miniaturization, parallel processing etc.) to order to have better "impedance matching" with the speed of Echo-MS analysis.

3. With the endogenous levels of the 10-plex protein biomarkers having such a wide dynamic range, was there an effort to optimize the SISCAPA antibody levels for these proteins?

4. The Echo-MS method has been characterized very thoroughly by the authors. One potential improvement would be to characterize the matrix effect (suppression effect) of individual plasma lot, as typically done in small molecule bioanalysis. Granted, the current method utilized SIL which would compensate for the matrix effect to a large extent. However the utilization of an intensity cut-off could potentially have a negative impact on samples from certain plasma lots if there're significant suppression.

5. In the Methods section, line 426, the correct term of "carrier solvent" was used for AEMS method description. However, in the supplemental information, the incorrect term of "mobile phase" was used instead.

Reviewer #3:

Remarks to the Author:

The authors describe application of an acoustic ejection mass spectrometry sampling method to measuring peptides from acute phase reactants in plasma and peptides from SARS-CoV-2 in nasal swabs. Studies were conducted in accordance to IRB review. The method is a nice demonstration of the potential for rapid analysis of biospecimens using the technology. I have the following minor recommendations to improve the technical clarity of the manuscript.

1. Figure 2B, y-axis label, should this be changed from "%CV Peak Area" to "%CV"? Since area ratio is also plotted on the same axes?

2. One validation aspect that I may have missed and could be a nice addition is the 19 technical

- replicates for the healthy pooled sample. Were these spread across multiple plates? If so, the distribution of CVs for that sample might provide additional between-plate imprecision.
3. On line 209 (and Figure 3C), results for 5 of 8 peptides are shown. Why were the other 3 eliminated?
 4. Line 220, Given an approximately 30-fold difference in sensitivity, I'm assuming the rejection rates were not the same for the LC-MS runs? How many data points were rejected in the LC-MS runs? How much did the difference in sensitivity matter for these analytes?
 5. Lines 231-239, please add p-values for statistical significant differences.
 6. If I understand correctly, line 292 states Ct value <40 is a positive sample, and good correlation with the AEMS results is found down to Ct25. Can the authors comment on the sample where viral peptides are detected in PCR-negative samples? Are these potential false negatives from the PCR test? Or is there another source of peptide signal present?
 7. In Methods section, what was the final volume of sample available for AEMS analysis (i.e., after peptide enrichment? Please clarify total sample consumed (e.g. 13 analytes X 300nL x triplicate = ~12 microliters?).
 8. In supplemental line 254 and 268 state different matrices used for the concentration curves, please clarify?
 9. Minor point, the utility of the PepCalMix Concentration curves is dubious. Curves were generated for PepCalMix (supp fig 5) and ARP analyte peptides (thresholds in supp fig 6) but only shown for PepCalMix. Some clarification may improve why the PepCalMix was important to the demonstration and help the readability of the manuscript.
 10. In Supp Fig 6, please distinguish which rejections are from insufficient peak area vs rejections from differences in fragment ratios.
 11. Supp Fig 8B, what is the difference in solid and patterned fills? Lines 329 and 330 says majority of rejection are in Ct>25 but the opposite is true in the figure - Are the Ct>25 and Ct<25 labels swapped (the way I'm reading it there should be more rejections at the highest Ct values)?
 12. Please add linear regression parameters to supp fig 10 (like the main figure).

Reviewer #1 (Remarks to the Author):

This manuscript presents a new method utilizing AEMS for the quantification of acute phase response protein markers and SARS-CoV-2 NCAP peptides in automated assays. The subject matter is relevant to the readership of Nature Communications, Nature Portfolio, and I endorse its publication pending the addressing of the following comments.

We extend our sincere appreciation to the reviewer for conducting a thorough examination of our work and providing valuable suggestions for improvement.

1. The article effectively optimizes the AEMS methods, addressing factors such as the repetitive readout of a particular plate for acute phase response proteins, requiring the plate to be read 10 times to analyze the entire set of 10 proteins, one protein at a time. Additionally, it optimizes parameters such as mobile phase composition, flow rate, sample ejection volume, dwell time, etc. However, a more detailed comparison of the LLOQ between AEMS and LC-MSMS is warranted. Firstly, the LLOQ of LC-MSMS is currently absent, and it was performed on the Sciex TQD system. Secondly, the previous result is not accurately presented in the paper. On Page 10, it states, "In a previous study using LC-MS, excellent linearity down to 4 amol/ μ L was observed, corresponding to 72 amol on column (9)." However, in line 99 of the previous publication (https://pubs.acs.org/doi/suppl/10.1021/acs.analchem.2c01610/suppl_file/ac2c01610_si_001.pdf), it was mentioned that 2 amol/ μ L corresponded to 72 amol on the column. Thirdly, the comparison of ejection amounts between AEMS and LC-MSMS should be corrected. On Page 10, it states, "This LC-MS method applies 10 μ L on column, whereas the ejection volume with AEMS was 300 nL, resulting in a 33.3-fold higher loading in LC-MS." However, the previous reference reports a 10 μ L injection with a Waters Xevo TQ-XS system (References #9), while the injection volume of the current Sciex LCMS system is 1 μ L for the inflammation panel samples and 2 μ L for the SARS-CoV-2 sample (Page 14). A thorough comparison of the Lower Limit of Quantification (LLOQ) between AEMS and LCMS is essential to assess whether AEMS can attain a sensitivity level suitable for the assays, particularly when using Mass Spectrometry systems from the same vendor, as opposed to comparisons across different vendors. In this context, there is a 3.3-6.6-fold higher loading in Sciex LCMS. Given this disparity, how does the LLOQ of LCMS compare to that of AEMS?

Response: We express our gratitude to the reviewer for identifying the discrepancy in the reported Lower Limit of Quantification (LLOQ). In response to this observation, we have now rectified the error in the manuscript by adjusting the value from 72 amol to 144 amol on column. For the injection volume on the Sciex LC-MS system however, another conversion factor got lost in the details of the manuscript. More specifically, the SPE purified nasopharyngeal (NP) SARS-CoV-2 samples were eluted in organic elution buffer (40% ACN in 0.1% FA), which is perfectly fine for AEMS, yet required a 1/4 dilution with mobile phase A for LC-MS analysis. In other words, the 2 μ L injections of the SARS-CoV-2 NP samples were done on 1/4 diluted samples compared to AEMS, which is the equivalent of 500 nL instead of 300 nL, i.e. a 1.66-fold higher loading with LC-MS. For the inflammation panel, this is the 3.3-fold change calculated by the reviewer. **We have clarified this in the online methods section (LC-MS analysis) of the manuscript.**

We apologize for not being able to provide a direct comparison of AEMS and LC-MS LLOQs. Unfortunately, we lack direct access to a high flow (500 μ L/min) LC system that can be coupled to a SCIEX QTRAP 6500+ instrument to measure a dilution series for direct LLOQ assessment. As recognized by the reviewer, while AEMS is a unique setup, exclusively coupled to SCIEX instruments (SCIEX QTRAP 6500+ and more recently the ZenoTOF 7600), comparing LLOQs is a challenging task because of the multitude of variables involved, encompassing not only different instrument models like a QTOF vs a triple quad, but equally different transition selection, differences in LC flow rates (500 μ L/min vs 5 μ L/min) and even batch effects. Therefore, only a direct replacement of an LC system by an AE module at the same MS instrument to measure the same samples would enable us to directly measure this difference in LLOQ, which lies outside the scope of this manuscript.

What is feasible and has been added to the manuscript, is a comparison of the overall SISCAPA peptide assay performance when measured through AEMS vs LC-MS. Specifically we assessed the assay performance based on (i) a recombinant protein dilution series in UTM medium and (ii) measurement of leftovers of the same patient samples that were already measured in the previous report (although using a slightly different bead-capture washing procedure). We consider these two "transferable" in terms of assay comparison. For the dilution series, the -now corrected - claim holds that a spike of 4 amol/ μ L in UTM could still be quantified using LC-MS (on a Xevo TQ-XS with high flow LC operating at 800 μ L/min), while AEMS can only quantify signal down to 195 amol/ μ L in UTM, which is rather a 48-fold loss in sensitivity over the entire assay, including the enrichment and injection volumes (which are not free to choose). When we turn to the 92/145 nasopharyngeal patient samples which are identical to the ones from the Cov²MS study and were analyzed here through AEMS, we can use an alternative quantitative ground truth,

i.e. their Ct value. These samples were measured two years ago on a Xevo TQ-XS (Waters Corporation) coupled to a high flow LC, in a different batch and in a different lab (Belgium). Here, we provide an “expandable” comparison by comparing assay performance (rather than LLOQ). Since we detected positive patients up to Ct 26 with AEMS and up to Ct 31 in the LC-MS Cov²MS measurements, we have proposed a 5 Ct difference in performance in this manuscript which translates to a 32-fold difference (which is comparable to the dilution series estimation). While this does not directly compare LLOQ values between LC-MS and AEMS, it does provide us with a reasonable estimation of sensitivity loss over the assay, i.e. roughly 40-fold. This is partially attributable to differences in sample load, i.e. a 1.6 to 3.3-fold, leaving a 10 to 20-fold difference attributable to the instrument setup.

While we want to refrain from direct LLOQ calculations in the text to avoid confusion, we did re-analyze the data from the LC-MS measurements done on a subset of these samples that were also analyzed by AEMS (L/H-ratio correlations between AEMS and LC-MS presented in **Figure 3C and Supplementary Figure 8 and 13, revised manuscript**). These 71 samples of the inflammation panel and 48 samples from the NP swabs were analyzed in a different batch on a different SCIEX QTRAP 6500+ instrument with a CapLC (10 μ L/min) system for sample loading. The peak areas, on average, were 10-20 fold lower for the AE data after correction of the injection volume described above, suggesting a number for the instrumental setup difference as expected. At least partially, this could be the combined effect of a 5-10 fold sensitivity difference in the LC flow rate and accompanying source settings (as described in the **Online Methods section, revised manuscript**) with a performance difference of the instrument itself. Assessing the impact of each of these variables lies outside the scope of this manuscript and we hope to have convinced the reviewer that our estimations for loss in assay performance make most sense in this report.

As this is not an analytically defined LLOQ, we added a section in the **Discussion on Study Limitation** to accommodate the reviewer's request: “However, this notable increase in throughput comes with an analytical cost, i.e. a lower sensitivity. While the manuscript currently lacks a direct LLOQ comparison with a comparable high-flow LC-MS approach, using (i) a dilution series of recombinant protein in UTM and (ii) the same patient samples and their known Ct values measured earlier (9) as a proxy, a roughly 40-fold reduction in sensitivity was found, which is partially due to differential sample loading amounts”.

2. The L/H ratios of acute phase response proteins within COVID-19 disease cohorts serve as severity biomarkers to analyze the positive/negative status of COVID-19. Regarding the discussion of Figure 4 on Page 9, has statistical analysis been conducted for the increase in CRP, LBP, SAA1:SAA2, and the decrease in Alb in COVID-19 positive samples? For a thorough analysis, it is advisable to conduct a paired t-test in both positive and negative sub-cohorts (e.g., COVID Positive, Admitted, death vs. COVID Negative, Admitted, death, and other relevant sub-cohorts).

Response: We thank the reviewer for the comment. As suggested, we performed unpaired t-tests for CRP, LBP, Alb and SAA between the Healthy and Diseased Samples, with corresponding p-values presented in the main text. We also performed several other comparisons, which indicate a potential upregulation of SAA levels in Covid positive samples compared to Covid negative samples. While the unpaired t-test yielded a statistically significant difference ($p < 0.0145$), it is noteworthy that employing a more stringent cutoff of 0.01 might have rendered the result statistically insignificant.

Therefore, we have included a statement in the manuscript (**Acute phase response proteins and Covid-19 disease severity section**) to emphasize the limitation of our discovery cohort's size in detecting subtle changes with adequate statistical confidence:

“As this is only a discovery cohort of 225 samples, other comparisons did not directly result in statistically significant differences. Moreover, it's worth highlighting that with the increased speed of data acquisition facilitated by AEMS, such comparative analyses comprising thousands of samples can now be performed more readily, potentially enhancing the precision of our findings in future studies.”

3. Figure 5 illustrates the comparison of samples analyzed by AEMS and RT-PCR. However, the accuracy evaluation of the SARS-CoV-2 NSAP assay with AEMS is not clearly elucidated for the 142 nasopharyngeal swab samples when using RT-PCR results as references. I am interested in understanding the percent positive agreement (PPA) and percent negative agreement (PNA), especially when $Ct < 25$ and $\text{Log}_2 \text{Area AYN} > 7.3$ are considered positive. Previous reports (References #8 and #9) demonstrate consistent results between RT-PCR and LCMS, both adhering to the CLSI EP 12-A2 User Protocol for Evaluation of Qualitative Test Performance guidance. How does the current LCMS data compare to RT-PCR? Through these comparisons, we can gain insights into the predictable accuracy for COVID-19 among AEMS, LCMS, and RT-PCR.

Response: We have addressed this concern by adding a PPA/PNA matrix in Figure 5 and **an additional Supplementary Figure S10**. Initially, we had refrained from this comparison for all the reasons given in the previously published Cov²MS manuscript. There, we did such comparison, yet added that “However,

there are several concerns with this representation. First, both tests report a continuous measure rather than a binary outcome and a threshold needs to be chosen to define “positive” and “negative”, arguably not trivial for either test. In fact, the positivity threshold for RT-qPCR varies greatly between assays and should probably be set to Ct 31 in light of recent insights on infectivity (1). Second, the numbers reported in the matrix are a direct function of the Ct distribution in the patient population tested. Supplementary Figure S10 illustrates that shifting, e.g., the positivity threshold of qPCR from Ct 40 to Ct 35 would not impact PPA or PNA here because there are no patient samples in that region. Third, RT-qPCR is not well suited to define the amount of RNA in copies/mL, with measurements sometimes differing >1000-fold between laboratories (2). MS on the other hand is considered a quantitative and accurate analytical tool, an asset typically not attributed to other protein detection technologies, such as lateral flow antigen tests (3). In other words, the Ct reported for a patient can vary greatly and thus defining RT-qPCR as the ground truth or golden standard does not objectify the comparison. Finally, there is a potential underlying biological reason why RNA and protein do not completely correlate, i.e., the stage of infection (4, 5). Indeed, while RNA and protein levels will most probably rise in parallel at the onset of infection, it is known that residual RNA can still be detected over a month following infection when the disease symptoms are no longer apparent (1,4,5).”

- (1) Bruce, E. A.; Mills, M. G.; Sampoleo, R.; Perchetti, G. A.; Huang, M.; Despres, H. W.; Schmidt, M. M.; Roychoudhury, P.; Shirley, D. J.; Jerome, K. R.; Greninger, A. L.; Botten, J. W. Predicting Infectivity: Comparing Four PCR-based Assays to Detect Culturable SARS-CoV-2 in Clinical Samples. *EMBO Mol. Med.* 2022, 14, e15290 DOI: 10.15252/EMMM.202115290
- (2) Evans, D.; Cowen, S.; Kammel, M.; O’Sullivan, D. M.; Stewart, G.; Grunert, H.-P.; Moran-Gilad, J.; Verwilt, J.; In, J.; Vandesompele, J.; Harris, K.; Hong, K. H.; Storey, N.; Hingley-Wilson, S.; Dühring, U.; Bae, Y.-K.; Foy, C. A.; Braybrook, J.; Zeichhardt, H.; Huggett, J. F. The Dangers of Using Cq to Quantify Nucleic Acid in Biological Samples: A Lesson From COVID-19. *Clin. Chem.* 2021, 68, 153– 162, DOI: 10.1093/clinchem/hvab219
- (3) Anderson, N. L.; Anderson, N. G.; Haines, L. R.; Hardie, D. B.; Olafson, R. W.; Pearson, T. W. Mass Spectrometric Quantitation of Peptides and Proteins Using Stable Isotope Standards and Capture by Anti-Peptide Antibodies (SISCAPA). *J. Proteome Res.* 2004, 3, 235– 244, DOI: 10.1021/PR034086H
- (4) Kissler, S. M.; Fauver, J. R.; Mack, C.; Olesen, S. W.; Tai, C.; Shiue, K. Y.; Kalinich, C. C.; Jednak, S.; Ott, I. M.; Vogels, C. B. F.; Wohlgemuth, J.; Weisberger, J.; DiFiori, J.; Anderson, D. J.; Mancell, J.; Ho, D. D.; Grubaugh, N. D.; Grad, Y. H. Viral Dynamics of Acute SARS-CoV-2 Infection and Applications to Diagnostic and Public Health Strategies. *PLoS Biol.* 2021, 19, e3001333 DOI: 10.1371/JOURNAL.PBIO.3001333
- (5) Mina, M. J.; Parker, R.; Larremore, D. B. Rethinking Covid-19 Test Sensitivity – A Strategy for Containment. *N. Engl. J. Med.* 2020, 383, e120 DOI: 10.1056/NEJMp2025631

Supplementary Figure 10. Ct distribution of nasopharyngeal swab samples. Bar graph illustrating the distribution of samples per Ct-value, showing the frequency of samples across different Ct ranges

Still, we agree with the reviewer that it is desirable for the readership to see such direct numerical comparison. We therefore opted to refer to the Cov2MS manuscript for the above-mentioned reasons for caution and did calculate the PPA/PNA. While plotting the numbers, we effectively found that best

performance is achieved at Log2AYN 7.5 and a Ct positivity threshold of 26 on this sample cohort. The PPA and PNA matrix was included in **Figure 5**. In addition, we added the following sentence in the section: *Quantification of SARS-CoV-2 NCAP peptide enriched from nasopharyngeal swabs*:

"In accordance to the Clinical and Laboratory Standards Institute (CLSI) user protocol for evaluation of qualitative test performance (EP 12-A2), percent positive and negative agreement (PPA and PNA) were calculated. The PNA/PPA matrix in Figure 5C depicts these numbers when a summed intensity of AYNVTQAFGR of 7.5 is used. However, there are several concerns with this representation as previously described by Van Puyvelde et al., therefore..."

		E gene PCR @Ct 26		
		Pos	Neg	
AYN Log2 @ 7.5	Pos	84	3	PPA = 91.30%
	Neg	8	46	PNA = 93.88%

4. Regarding the Solid-Phase Extraction (SPE) clean-up for peptide enrichment, the elution of the peptide involved 2x10 μ L of 40% acetonitrile in 0.1% formic acid, followed by dilution with 35 μ L of a 0.5% formic acid solution in H₂O. Has there been an assessment through recovery experiments to confirm the accuracy and reproducibility of this step? Typically, after eluting the analyte from the SPE column, the common practice is to evaporate the solution and reconstitute it with a mobile phase solution or another buffer to maintain a consistent volume in the final samples. This step is essential because elution volumes often exhibit inconsistency in SPE, regardless of whether a positive pressure or vacuum procedure is utilized. I understand that resolving the peptides by drying down and reconstituting may introduce additional challenges.

Response: We acknowledge the reviewer's concern regarding potential inconsistencies in elution volumes. We had anticipated this concern and implemented stable isotopic peptides (SIL with N15) to allow for comparison to the N13 endogenous peptide as a robust control mechanism. Since SPE was exclusively applied to the nasopharyngeal swab samples, we felt like it might overly complicate the manuscript if we address it in more detail. Moreover, the decision to forgo a drying step in the sample preparation process was deliberate, as we aimed to streamline the sample preparation to make it as high-throughput as possible. Drying down the eluted peptides could introduce further intricacies to the workflow. Also note that with AEMS, there is not the same requirement to reduce the organic composition of a sample post-SPE as with LC-MS analysis, so we opted for using a simpler dilution of the sample for AEMS rather than a dry down. We trust that the reviewer understands and agrees with our rationale for not elaborating on these aspects, as outlined in our response. If there are any specific concerns or suggestions for improvement, we welcome further discussion and input.

5. Furthermore, there is a need for improvement in the legend and axis units of Supplementary Figure 3. It is challenging to determine the accuracy of CHAPS and 1xPBS as the maximum tolerated concentration is presented in a different unit on the axis, which I presume to be mM. In the case of TRIZMA, the maximum tolerated concentration appears to be around 0.0006 mM, assuming a similar interpretation as the other two additives, NaCl and AmBic.

Response: Thank you for pointing out this unclarity. We have now added the starting concentrations in **Supplementary Figure 3B** and the corresponding caption. In addition, we have included a sentence in **Supplementary Material** to clarify the starting concentrations: *"The starting concentration for each of the tested compounds in the 1:2 dilution series are 0.03% CHAPS, 1x PBS, 100 mM NaCl, 0.2M TRIZMA and 100 mM ABC."*

Reviewer #2 (Remarks to the Author):

This manuscript describes an Echo-MS method for high-throughput peptide quantification after sample enrichment using immunocapture from digested biological matrices. This is the first report of using Echo-MS technology for peptide analysis, and the authors have demonstrated excellent quality and

reproducibility of the methodology. The manuscript could be further improved by addressing the following comments.

We would like to thank the reviewer for the appreciation of our work and for the valuable suggestions provided to improve the manuscript.

1. With a limit of 4 MRM channels per Echo-MS method, it seemed that each sample had to be analyzed multiple times, once for each H/L peptide pair. Does that mean for the 10-plex SISCAPA extracted samples, each sample was analyzed 30 times (3 technical replicates x 10 peptide pairs)? The need for multiple Echo-MS analyses on the same sample was mentioned in the manuscript, but I think it would need to be clarified in the throughput discussion (page 8, line 215).

Response: Indeed, the samples were analyzed in triplicate as a proof of principle and in order to assess the quantitative accuracy of the measurement. Therefore, we now more clearly state in the manuscript (Section: Acute phase response proteins enriched from plasma samples – data quality) that based on our findings; **single measurements should be adequate when AEMS is applied in the future.**

“Nevertheless, the robust reproducibility observed in the SIL peptide area (Figure 3B, inset) serves as evidence that triplicate measurements are dispensable. This revelation presents a noteworthy prospect for a substantial reduction in analysis time, allowing for the completion of a single 96-well plate in just 32 minutes.”

2. With a throughput of ~5 hrs/96-well plate on Echo-MS and 1 day/96-well for sample preparation, there's an obvious mismatch and the bottleneck was shifted to the automated sample preparation step. I think it would be helpful to add some discussion on how to further improve the speed of digestion and immunocapture (miniaturization, parallel processing etc.) to order to have better “impedance matching” with the speed of Echo-MS analysis.

Response: We concur with the reviewer's observation that the bottleneck has transitioned from sample analysis to sample preparation. In response to this, we have emphasized the importance of automation in our expanded **discussion section and provide potential strategies** to enhance the speed of the digestion process. An additional reference on automated proteomics sample preparation was also cited in the revised manuscript (Reference #39).

Line 340-345: "While it was not tested here, it is feasible to decrease trypsin incubation to a 30 minute interval, or to perform this step offline to reciprocally increase the number of 96-well plates that can be processed daily by a single operator (12). Using automation stations equipped with multiple shakers and incubators, processing two 96 well plates in 4.5 hours, or ten 96-well plates per day is possible (25). Therefore, with more advanced automation stations, fully automated sample preparation is achievable, and coupling the Echo MS system with automatic plate loading systems can enable continuous 24/7 operation (39)."

3. With the endogenous levels of the 10-plex protein biomarkers having such a wide dynamic range, was there an effort to optimize the SISCAPA antibody levels for these proteins?

Response: This an insightful question. The acute phase response (APR) protein 10-plex antibody mixture was procured directly from SISCAPA Assay Technologies, who have previously conducted optimization of antibody concentration levels based on the dynamic range as outlined in Razavi et al. (2016). For clarity we have modified the Online Methods - Reagents and materials section.

4. The Echo-MS method has been characterized very thoroughly by the authors. One potential improvement would be to characterize the matrix effect (suppression effect) of individual plasma lot, as typically done in small molecule bioanalysis. Granted, the current method utilized SIL which would compensate for the matrix effect to a large extent. However the utilization of an intensity cut-off could potentially have a negative impact on samples from certain plasma lots if there're significant suppression.

Response: We thank the reviewer for the comment. While we acknowledge that employing an intensity cut-off may pose potential implications for samples from specific plasma lots experiencing significant suppression, it's important to note that this concern transcends AEMS and is inherent to MS methodologies in general. Nevertheless, given that our samples undergo extensive purification via immunocapture combined with four thorough bead washing steps, we do not anticipate significant performance discrepancies among various plasma pools. Even in the unlikely event of such discrepancies, the incorporation of the heavy spike-ins will aid us in addressing this issue effectively. In the Online Methods - Peptide enrichment section we have emphasized that with SISCAPA, the analytes undergo extensive washing to reduce background which could cause performance discrepancies between plasma pools or samples.

5. In the Methods section, line 426, the correct term of "carrier solvent" was used for AEMS method description. However, in the supplemental information, the incorrect term of "mobile phase" was used instead.

Response: Well caught, we have adjusted this in the manuscript.

Reviewer #3 (Remarks to the Author):

The authors describe application of an acoustic ejection mass spectrometry sampling method to measuring peptides from acute phase reactants in plasma and peptides from SARS-CoV-2 in nasal swabs. Studies were conducted in accordance to IRB review. The method is a nice demonstration of the potential for rapid analysis of biospecimens using the technology. I have the following minor recommendations to improve the technical clarity of the manuscript.

We express our gratitude to the reviewer for the meticulous feedback, particularly on the supplementary figures. These insightful suggestions have enhanced the clarity and overall quality of the manuscript.

1. Figure 2B, y-axis label, should this be changed from "%CV Peak Area" to "%CV"? Since area ratio is also plotted on the same axes?

Response: Thank you and we have **adjusted Figure 2B** accordingly.

2. One validation aspect that I may have missed and could be a nice addition is the 19 technical replicates for the healthy pooled sample. Were these spread across multiple plates? If so, the distribution of CVs for that sample might provide additional between-plate imprecision.

Response: We thank the reviewer for the suggestion. The plasma cohort was fully randomized and included several replicates of the healthy pooled JVE plasma (annotated as JVE_P) across the three sample plates to assess the reproducibility of the total workflow (sample preparation + Echo MS analysis). For clarity, we have added the **plate layouts** in our response. We have added an **additional supplementary Figure 6** and **report on the reproducibility** of the L/H ratio sum from the 19 pooled JVE Plasma samples in the main text.

"The %CV for the L/H-ratio sum for the 10 APR proteins ranged from 6.68% to 40.87% (Supplementary Fig. 6)."

	1	2	3	4	5	6	7	8	9	10	11	12	
A	JVE_P	763	638	1886	1003	462	374	2208	1124	JVE_P	1085	2595	A
B	377	613	573	1264	1301	787	JVE_P	2476	1120	1118	1995	1209	B
C	320	JVE_P	1355	345	1002	2071	484	JVE_P	496	1285	401	639	C
D	1026	1006	1011	1004	1096	1018	1024	563	JVE_P	1005	408	400	D
E	574	1010	463	JVE_P	1007	JVE_P	2608	1276	2436	1152	1283	658	E
F	1013	578	JVE_P	2104	1164	1278	698	1135	760	461	606	JVE_P	F
G	414	2807	2956	2627	493	1168	475	742	2692	2519	2481	360	G
H	405	2919	1064	1814	JVE_P	706	736	582	725	1128	JVE_P	873	H
	1	2	3	4	5	6	7	8	9	10	11	12	

Plate Layout - Plate 1

	1	2	3	4	5	6	7	8	9	10	11	12	
A	JVE_P	1384	1784	1082	1058	2486	399	621	JV35	1887	1055	JVEP_P - 0 pmol	A
B	1415	860	JV29	1065	1347	JV20	1091	JV38	420	887	1455	BvP baseline	B
C	1253	577	573	JV32	625	1394	1236	1359	2390	1068	778	BvP 7d postCVD	C
D	659	692	695	1436	1078	702	2586	JVE_P	1774	JV16	JV30	BvP 13d postCVD	D
E	JV39	398	1462	JVE_P	757	417	JV8	389	1993	1580	1772	BvP 20d postCVD	E
F	1249	JV12	JV9	1404	380	343	1343	416	781	768	1798	BvP 27d postCVD	F
G	1880	2632	1153	1073	1063	2181	694	1790	434	1123	JVE_P - 4.19 fmol	JVE_P - 10.4 fmol	G
H	JVE_P - 26.2 fmol	JVE_P - 65.5 fmol	JVE_P - 163.84 fmol	JVE_P - 409.6 fmol	JVE_P - 1.024 pmol	JVE_P - 2.56 pmol	JVE_P - 6.4 pmol	JVE_P - 16 pmol	JVE_P - 40 pmol	JVE_P - 100 pmol	JVE_P - 400 pmol	JVE_P - 1000 pmol	H
	1	2	3	4	5	6	7	8	9	10	11	12	

Plate Layout - Plate 2

	1	2	3	4	5	6	7	8	9	10	11	12	
A	JVE_P	1898	1391	1092	1583	1093	1076	455	1458	2134	699	JV36	A
B	2440	1181	1933	1198	1363	1140	1482	1463	1089	1781	1575	2689	B
C	2385	JV2	1345	JVE_P	JV28	1021	442	2492	1174	456	1351	2478	C
D	1680	1938	1180	1366	619	1916	1799	2479	1248	1438	JV18	1905	D
E	344	1568	JV15	1479	421	JV23	1798	JVE_P	JV37	2416	1782	2370	E
F	2382	1374	1779	JV19	2237	1179	JV6	2904	1674	1457	1926	1388	F
G	1432	1170	2505	1017	2235	1430	1187	JV11	693	JV22	1686	JV3	G
H	JV17	373	2951	1176	2428	1884	1390	1579	1469	1348	1408	JVE_P	H
	1	2	3	4	5	6	7	8	9	10	11	12	

Plate Layout - Plate 3

Supplementary Figure 6. Reproducibility of replicate sample processing of a plasma pool. Variability of L/H ratios across 19 replicate wells of pooled healthy plasma for all ten APR peptides across the three sample plates.

3. On line 209 (and Figure 3C), results for 5 of 8 peptides are shown. Why were the other 3 eliminated?

Response: Indeed, we only displayed the most biologically relevant biomarker proteins, namely A1AG, C3, LBP, CRP and SAA, these have also been reported in earlier studies as being potentially relevant in the context of a SARS-CoV-2 infection. We have now **included the L/H ratio** correlations for the other three proteins (Alb, Hx and IgM) in **Supplementary Figure 8**.

Supplementary Figure 8. Correlation of measured AEMS L/H peptide ratios with LC-MS data. The ratios measured for Alb, Hx and IgM by LC-MS were very similar to the ratios determined using the Echo MS system.

4. Line 220, Given an approximately 30-fold difference in sensitivity, I'm assuming the rejection rates were not the same for the LC-MS runs? How many data points were rejected in the LC-MS runs? How much did the difference in sensitivity matter for these analytes?

Response: This is a good question and your assumption is correct that the LC-MS sensitivity was higher, as we ran microflow LC (10 μ L/min) on the SCIEX 6500+ system with higher injection volumes to obtain L/H ratios for comparison to the Echo MS data (which is operating at 500 μ L/min). The same rejection strategy was applied to the LC-MS data, including using the same ratio standard deviation and same peak area thresholds (which we feel will be conservative for the LC-MS data as the peak area is \sim 15x higher or more depending on the peptide). Interestingly, similar rejection rates were seen for most proteins driven mainly by the ratio standard deviation rejection. For the low abundant proteins, i.e. SAA, MPO and MBL, the rejection rates were substantially lower for the LC-MS data because more data points exceeded the intensity threshold. This rejection data was added to **Supplementary Figure 7 as an additional bar on the graph**.

5. Lines 231-239, please add p-values for statistical significant differences.

Response: We thank the reviewer for this comment, especially since a similar concern was raised by reviewer 1.

As suggested, we performed unpaired t-tests for CRP, LBP, Alb and SAA between Healthy and Diseased Samples, with corresponding p-values presented in the main text. We also performed several other comparisons, which indicate a potential upregulation of SAA levels in Covid positive samples compared to Covid negative samples. While the unpaired t-test yielded a statistically significant difference ($p < 0.0145$), it is noteworthy that employing a more stringent cutoff of 0.01 might have rendered the result statistically insignificant.

Therefore, we have included a statement in the manuscript (**Acute phase response proteins and Covid-19 disease severity section**) to emphasize the limitation of our discovery cohort's size in detecting subtle changes with adequate statistical confidence:

"As this is only a discovery cohort of 225 samples, other comparisons did not directly result in statistically significant differences. Moreover, it's worth highlighting that with the increased speed of data acquisition facilitated by AEMS, such comparative analyses comprising thousands of samples can now be performed more readily, potentially enhancing the precision of our findings in future studies."

6.If I understand correctly, line 292 states Ct value <40 is a positive sample, and good correlation with the AEMS results is found down to Ct 25. Can the authors comment on the sample where viral peptides are detected in PCR-negative samples? Are these potential false negatives from the PCR test? Or is there another source of peptide signal present?

Response: We have now **included a PPA/PNA matrix as part of Figure 5** as requested by reviewer 1. In doing so, we found that the assay has the greatest agreement with RT-qPCR with a Log₂ Peak Area AYNVTQAFGR (Log₂AYN) at 7.5 as a positivity threshold at Ct 26 for RT-qPCR. Still, following this adjustment, several PCR-negative samples are considered positive in the assumption of Log₂ Peak Area of 7.5. Most strikingly, some of these samples also have signal for a second peptide i.e. ADE. A more elaborate argumentation in the pitfalls of using RT-qPCR as a golden standard can be found in the Cov²MS manuscript and under our response to reviewer 1 – remark 3.

7.In Methods section, what was the final volume of sample available for AEMS analysis (i.e., after peptide enrichment? Please clarify total sample consumed (e.g. 13 analytes X 300nL x triplicate = ~12 microliters?).

Response: This is a thoughtful comment. We have now included the final volume available after SISCAPA enrichment, i.e. 50 μ L, in the Online Methods – Peptide enrichment section. However, only 30 μ L can be used for AEMS analysis since a minimum liquid height is required in sample well for optimal acoustic droplet ejection. In the case of a 384-well plate, a minimum volume of 20 μ L must be retained in each well, this leaves 30 μ L available from our total 50 μ L for AEMS analysis, allowing for the execution of up to 100 ejections. Additionally, even in a 1536-well plate scenario (minimum liquid height is 2 μ L), where only 3 μ L out of the initial 7 μ L is required for efficient ejection, a comprehensive analysis of all 10 APR proteins can be conducted in a single measurement. This important detail has been incorporated into the Online Methods – AEMS analysis section of the revised manuscript. It is worth noting that triplicate measurements are no longer a prerequisite in the final implementation due to the technical accuracy of the Echo-MS system. The revised protocol specifies that a minimal sample volume for 10 ejections, equivalent to 3 μ L, is sufficient to screen the complete APR panel.

Please note that given the demonstrated reproducibility of peptide quantification with AEMS, future studies will likely opt for single replicates, resulting in a one-third reduction in volume consumption. Therefore, we now more clearly state in the manuscript (Section: Acute phase response proteins enriched from plasma samples – data quality) that based on our findings; **single measurements should be adequate when AEMS is applied in the future.**

"Nevertheless, the robust reproducibility observed in the SIL peptide area (Figure 3B, inset) serves as evidence that triplicate measurements are dispensable. This revelation presents a noteworthy prospect for a substantial reduction in analysis time, allowing for the completion of a single 96-well plate in just 32 minutes."

8.In supplemental line 254 and 268 state different matrices used for the concentration curves, please clarify?

Response: Thank you for bringing this to our attention and we apologize for any confusion caused and appreciate your diligence in reviewing our work. Indeed, there is a discrepancy in the matrices mentioned

for concentration curves at supplemental lines 254 and 268. At some point, we have measured a dilution series of the PepCalMix both in a simple matrix and in a bit more complex background being a beta-galactosidase digest. Since there was no difference in performance, we finally decided to only report the calibration curves in the simple matrix (0.5% FA, 0.00025% CHAPS). We have rectified this inconsistency in the caption of Supplementary Figure 5.

9. Minor point, the utility of the PepCalMix Concentration curves is dubious. Curves were generated for PepCalMix (supp fig 5) and ARP analyte peptides (thresholds in supp fig 6) but only shown for PepCalMix. Some clarification may improve why the PepCalMix was important to the demonstration and help the readability of the manuscript.

Response: The PepCalMix kit contains a pooled mixture of 20 synthetic peptides which provide coverage over a wide range of masses and peptide properties. Our hope is that PepCalMix, can become a sensitivity test for researchers to adopt Echo MS for peptide quantitation, especially since we also made dilution series for APR, we now also **report on those as well (see Supplementary Table 7)**. Overall, the manuscript now more clearly describes both peptide mixtures used.

Supplementary Table 7. Lower limits of quantification (LLOQs) for 10 APR peptides using AEMS. Lower limits of quantification were determined as the lowest concentration where the %CV across 5 replicates was < 20% and the accuracy was between 80-120%.

Peptide	Instrument 1			
	LLOQ (fmol/uL)	%CV	%Accuracy	R2
A1AG.NWGLSVYADKPETTK	0.610	10.3	118.3	0.991
Alb.LVNEVTEFAK	0.310	17.5	89.4	0.993
C3.IHWESASLLR	1.220	18.1	118.8	0.997
CRP.ESDTSYVSLK	0.610	9.2	87.2	0.998
Hx.NFPSPVDAEFR	0.150	14.3	96.1	0.996
IgM.YAATSQVLLPSK	0.610	21.8	81.5	0.998
LBP.LAEGFPLPLLK	0.610	9.7	106.9	0.989
MBL.EEAFLGITDEK	0.310	14.8	84.7	0.995
MPO.DYLPLVLGPTAMR	0.610	8.4	107.3	0.992
SAA.GPGGVWAAEAISDAR	0.610	4.5	129.9	0.994
Average	0.57	12.85	102.00	0.994

10. In Supp Fig 6, please distinguish which rejections are from insufficient peak area vs rejections from differences in fragment ratios.

Response: We thank the reviewer for this comment. We have now added this information to the manuscript by adding an **additional bar to the graph in Supplementary Figure 7**. So both the total LC-MS rejections and the Ratio standard deviation rejections are now presented in the graph.

11. Supp Fig 8B, what is the difference in solid and patterned fills? Lines 329 and 330 says majority of rejection are in Ct>25 but the opposite is true in the figure - Are the Ct>25 and Ct<25 labels swapped (the way I'm reading it there should be more rejections at the highest Ct values)?

Response: We express our gratitude to the reviewer for bringing attention to this matter, signaling a potential clarity issue in the Figure and/or Figure caption. The visualized data represents the total number of rejected data points (depicted by the filled pattern) within each group. For instance, we have 23 nasopharyngeal patient samples with Ct-values ranging from 25 to 40 and a majority of them are rejected. To enhance understanding, we have made adjustments to the figure axis and figure caption to provide a clearer interpretation of the displayed information.

A

Protein	Light peptide peak area minimum threshold	Heavy peptide peak area minimum threshold
SARSCoV2.AYNVTQAFGR	1000	1000
SARSCoV2.ADETQALPQR	1500	1500
SARSCoV2.KQQTVTLLPAADLDDFSK	1500	1500

B

12. Please add linear regression parameters to supp fig 10 (like the main figure).

Response: We thank the reviewer for this suggestion. We have now added the linear regression parameters to Supplementary Figure 13.

Reviewers' Comments:

Reviewer #1:

Remarks to the Author:

The findings of this study are notably significant, illustrating substantial progress through the utilization of the front-cut instrument AEMS. These results offer promising prospects for the field and its related areas, indicating potential pathways for further exploration and practical application. I extend my appreciation to the authors for their thorough response to all my inquiries. The clarifications made based on scientific observations and facts are comprehensible and acceptable. The conclusions drawn from the study are well-founded upon the data and analysis provided. In summary, this research exemplifies commendable dedication and considerable impact. The revised manuscript stands as a valuable addition to the realm of Acoustic Ejection Mass Spectrometry applications and warrants serious consideration for publication.

Reviewer #2:

Remarks to the Author:

The revised manuscript is ready for publication

Reviewer #3:

Remarks to the Author:

The authors have adequately addressed comments and the manuscript is suitable for publication.